# DATA SUBSET SELECTION VIA MACHINE TEACHING

## ABSTRACT

We study the problem of data subset selection: given a fully labeled dataset and a training procedure, select a subset such that training on that subset yields approximately the same test performance as training on the full dataset. We propose an algorithm, inspired by recent work in machine teaching, that has theoretical guarantees, compelling empirical performance, and is model-agnostic meaning the algorithm's only information comes from the predictions of models trained on subsets. Furthermore, we prove lower bounds that show that our algorithm achieves a subset with near-optimal size (under computational hardness assumptions) while training on a number of subsets that is optimal up to extraneous log factors. We then empirically compare our algorithm, machine teaching algorithms, and coreset techniques on six common image datasets with convolutional neural networks. We find that our machine teaching algorithm can find a subset of CIFAR10 of size less than 16k that yields the same performance (5-6% error) as training on the full dataset of size 50k.

## 1 INTRODUCTION

Machine learning has made tremendous progress in the past two years with large models such as GPT-3 and CLIP (Brown et al., 2020; Radford et al., 2021). A key ingredient in these milestones is increasing the amount of training data to Internet scale. However, large datasets come with a cost, and not all – or maybe not even most – training data is useful. *Data subset selection* addresses this practical issue with the goal of finding a subset of a dataset so that training on the subset yields approximately the same test performance as training on the full dataset (Wei et al., 2015). Applications of data subset selection include continual learning (Aljundi et al., 2019; Borsos et al., 2020; Yoon et al., 2021), experience replay (Schaul et al., 2016; Hu et al., 2021), curriculum learning (Bengio et al., 2009; Wang et al., 2021), and data-efficient learning (Killamsetty et al., 2021a). Additionally, data subset selection is closely related to active learning (Settles, 2009; Sener & Savarese, 2018) and to fundamental questions about the role of data in learning (Toneva et al., 2018).

Data subset selection has been studied in two quite different contexts. First, perhaps the more popular approaches are found in the *coreset* literature (Sener & Savarese, 2018; Coleman et al., 2020; Mirzasoleiman et al., 2020; Paul et al., 2021), an empirically driven (Guo et al., 2022) research area that includes a variety of techniques to select important and diverse points, often with an eye towards minimal computational cost of subset selection. Second, *machine teaching* (Goldman & Kearns, 1995; Shinohara & Miyano, 1991) focuses on minimizing the number of examples a teacher must present to a learner and can be reframed as data subset selection in some settings. As machine teaching has mostly been studied from a conceptual or theoretical viewpoint, black-box models such as modern neural networks present a practical challenge for machine teaching. However, recently, a theoretical breakthrough in the machine teaching literature (Dasgupta et al., 2019; Cicalese et al., 2020) formalizes and provides algorithms with analysis for teaching black-box learners. Although these works are mainly theoretical, they also include some limited empirical evaluation, though with implementations that include unjustified heuristics. Furthermore, Dasgupta et al. (2019) includes no baselines and Cicalese et al. (2020) only includes random sampling as a baseline. As of yet, the two algorithms have not been compared in the literature, much less to other coreset methods.

In this work, we bring together these two lines of research, through both theoretical analysis and empirical evaluation. We make a clear connection between a particular machine teaching setting and data subset selection, use this insight to introduce an algorithm with state of the art and near-optimal asymptotics, prove correctness of implementation heuristics from previous machine teaching work,

and empirically evaluate methods from both machine teaching and coreset selection on a standard set of benchmarks for the first time. The subset size returned by our machine teaching algorithm shaves off a factor logarithmic in the dataset size compared to existing work. Furthermore, through novel lower bounds, we show that the subset size of our algorithm is near-optimal (under computational hardness assumptions regarding the NP-complete class of problems) and that the expected number of times we must query the learner (train a network) is optimal up to extraneous log factors.

Existing machine teaching algorithms from Dasgupta et al. (2019); Cicalese et al. (2020) perform well when our learner fits labels perfectly (zero training error) but fail catastrophically if the learner makes a few training errors. To address this issue, prior work (Cicalese et al., 2020) removes the training errors from the predictions provided to the teacher, a technique we refer to as *error squashing*. We provide a rigorous theoretical framework that explains why this technique of squashing errors is justified and effective, and furthermore, use it in our algorithm.

Finally, and perhaps most importantly, we compare machine teaching algorithms, including ours, to state-of-the-art coreset selection techniques and random sampling. We perform experiments with three convolutional neural network architectures on six image datasets (CIFAR10, CIFAR100, CINIC10, MNIST, Fashion MNIST, SVHN). We find that the machine teaching algorithms all perform roughly the same and consistently match or outperform the coreset selection techniques.

In summary, our main contributions are:

- Proposing a machine teaching subset selection algorithm with analysis and lower bounds, showing that the algorithm achieves optimal asymptotic performance up to extraneous log factors.

- Providing the first analysis for the justification of error squashing.

- Experimentally comparing machine teaching algorithms and coreset techniques using three convolutional neural network architectures on six image datasets.

In Sections 2 and 3, we introduce the classification setting and present our algorithm with its guarantee. Next, we show our main theoretical results in Section 4 and experimental results in Section 5. Finally, we discuss our work within the context of related work in Sections 6 and 7.

## 2 SETTING

We work in a classification setting with an input space $\mathcal{X}$ and a finite output space $\mathcal{Y}$. Given a distribution $\mathcal{D}$ over $\mathcal{X} \times \mathcal{Y}$, we wish to find a classifier $h : \mathcal{X} \to \mathcal{Y}$ with low (test) error: $\text{err}(h) = \Pr_{(x,y)\sim\mathcal{D}}[h(x) \neq y]$.

In this work, we assume we have a hypothesis class $\mathcal{H}$ and a learner $L$, where each $h \in \mathcal{H}$ is a function $h : \mathcal{X} \to \mathcal{Y}$ and the learner is a function: $L : (\mathcal{X} \times \mathcal{Y})^* \to \mathcal{H}$.

We assume we have a pool of $m$ labeled data points, $\{(x_i, y_i)\}_{i=1}^m \subset (\mathcal{X} \times \mathcal{Y})^m$. The objective of data subset selection is to select a subset $S \subset [m]$ of small size $|S|$ such that $\text{err}(L(\{(x_i, y_i) : i \in S\}))$ is low; perhaps as low as the error of training on all data: $\text{err}(L(\{(x_i, y_i) : i \in [m]\}))$.

## 3 METHOD

Although our ultimate objective is low test error, without a comprehensive understanding of generalization for models such as neural networks, we instrumentally focus on achieving low error on the pool of $m$ datapoints. Zhang et al. (2021) show that modern neural networks can exactly fit arbitrary (even random) ground truth labels. Following previous machine teaching work, we initially assume that the learner makes no errors on the training subset, and refer to these learners as *interpolating learners*. Note that for interpolating learners, we can always achieve *zero* pool error if we train with the entire pool. However, it may be possible to select a smaller subset that yields zero pool error. Later, we relax this assumption as it is only approximately true in practice.

### 3.1 MAIN ALGORITHM

In this section, we present our algorithm (see Algorithm 1 for pseudo-code). Our algorithm is an iterative algorithm, where each iteration is composed of three steps: sampling a subset of the pool, training on that subset, and using the results of training to update the subset sampling distribution. The intuitive principle behind our algorithm is that if we train on a subset $S \subset [m]$ and receive a hypothesis $h \in \mathcal{H}$, we want to emphasize the points (i.e., include some of the points in future subsets) where the hypothesis makes errors with respect to the ground truth labels. Then, in future iterations, that same hypothesis will not be returned by an interpolating learner since the hypothesis would make an error on the subset.

At each iteration, the algorithm samples a subset $S_t$ and trains on that subset to yield a hypothesis $h_t$. For sampling a subset at the $t^{th}$ iteration, the algorithm independently samples Bernoulli random variables with probability $p_{t,i}$, and upon success, includes the $i^{th}$ point. The success probability of the Bernoulli variable for the $i^{th}$ point at the $t^{th}$ iteration is $p_{t,i}$ and we sample a set $S_t$ such that $\Pr(S_t = S) = \left(\prod_{i \in S} p_{t,i}\right)\left(\prod_{i \in [m] \setminus S}(1 - p_{t,i})\right)$. Thus, in order to emphasize errors, the algorithm sets the next iteration's Bernoulli to have higher success probability ($p_{t+1,i} > p_{t,i}$).

While there are a variety of ways to increase the probabilities, our algorithm calculates the errors $E_t := E_{h_t}$ (where $E_h = \{i \in [m] : h(x_i) \neq y_i\}$) on the entire pool made by the hypothesis $h_t$ and sequentially doubles the probabilities corresponding to the errors until $\sum_{i \in E_t} p_{t,i}$ is sufficiently large (larger than a hyperparameter, $\xi$). Since a probability might exceed 1 after this step, we subsequently clip the probabilities to 1.

Because the algorithm only updates probabilities via doubling, a natural choice is to initialize $p_{1,i}$ as a (negative) power of 2: $2^{-k_0}$ for some integer $k_0$. Then, $p_{t,i}$ is always a power of two, since doubled powers of 2 are powers of 2.

The final aspect of the algorithm is that we periodically halve all probabilities. This step is key to improving the guarantee on the size of the subset from previous work (Dasgupta et al., 2019; Cicalese et al., 2020) to the asymptotically near-optimal subset size in this work. Note that without this halving step, the probabilities only increase with the number of iterations.

Our algorithm has two hyperparameters, $\hat{d}$ and $\hat{N}$, which are used to set the doubling limit $\xi$, the initial power $k_0$, and the number of iterations between halving the probabilities. The algorithm will succeed with high probability if the two hyperparameters are upper bounds on $d^*$, the size of the smallest subset that yields zero pool errors, and $N$, the size of the *induced hypothesis set*, a set we will define in the next section. Conveniently, in the cases where the algorithm fails, it returns whether $\hat{d}$ was too small or $\hat{N}$ was too small. We can set $\hat{d}$ and $\hat{N}$ to be arbitrarily large to yield valid subsets, but the size of the resulting subsets will be very large. See Algorithm 1 for pseudo-code.

### 3.2 ALGORITHMIC ANALYSIS

For the analysis, we begin by assuming that our learner is interpolating, meaning that it makes no errors on the subset it is trained on. More precisely, if $h_S = L(\{(x_i, y_i) : i \in S\})$ where $S \subset [m]$, then $\forall i \in S : h_S(x_i) = y_i$. Note that this implies that if we train on the entire dataset, a hypothesis that makes zero pool errors is returned.

Define $\overline{\mathcal{H}}$ as the induced hypothesis class, that is, all hypotheses that can be returned by training on a subset: $\overline{\mathcal{H}} = \{L(\{(x_i, y_i) : i \in S\}) : S \subset [m]\}$. Note that the size of $\overline{\mathcal{H}}$ could be as large as $2^m$ (the number of subsets) but can be much smaller if the data has structure that can be leveraged by the model. Our framework technically requires a deterministic learner and we can fix the random seed for training; however, this is not a practical issue as it is extremely unlikely that we will train a model on the exact same subset at two different iterations.

We now define $d^*$ and $N$. Let $E_h$ be the errors of a hypothesis $h$: $E_h = \{i \in [m] : h(x_i) \neq y_i\}$. Let $\mathcal{E}$ be the set of possible errors: $\mathcal{E} = \{E_h : h \in \overline{\mathcal{H}}\} - \{\emptyset\}$. We say a set $S$ "fully intersects" $\mathcal{E}$ if for all $E_h \in \mathcal{E}$, $|S \cap E_h| \geq 1$. Note that an interpolating learner trained on a fully intersecting subset $S$ will yield 0 pool error; if not, then the returned hypothesis $h_t$ would have non-zero training error on the subset $S_t$ which contradicts the interpolating assumption. Let $d^*$ be the size of smallest

---

**Algorithm 1**

---

**Input**: examples $\{(x_i, y_i)\}_{i=1}^m \subset (\mathcal{X} \times \mathcal{Y})^m$, failure probability $\delta$, $\hat{d} \in \mathbb{Z}_+$, $\hat{N} \in \mathbb{Z}_+$
$k_0 = \lfloor \log_2(m/\hat{d}) \rfloor$
$\xi = \ln(4\hat{N}^2 \log_2(m)/\delta)$
$\forall i : k_{1,i} = k_0$
**for** $t = 1, 2, \ldots, 4\hat{d}k_0$ **do**
    Sample $S_t$ such that $i \in S_t$ independently with probability $p_{t,i} = 2^{-k_{t,i}}$
    Train $h_t = L(\{(x_i, y_i)\}_{i \in S_t})$
    Calculate the errors of $h_t$ on the pool: $E_t := E_{h_t} = \{i \in [m] : h_t(x_i) \neq y_i\}$
    **if** $|E_t| = 0$ **then**
        **return SUCCESS:** $\hat{S} = S_t$
    **else**
        **if** $\sum_{i \in E_t} p_{t,i} \geq \xi$ **then**
            **return FAILURE TYPE 1:** $\hat{N}$ is too small
        **else**
            $\Delta_t = \left\lceil \log_2 \left( \frac{\xi}{\sum_{i \in E_t} p_{t,i}} \right) \right\rceil$
            $k_{t+1,i} = \max(0, k_{t,i} - \Delta_t \mathbf{1}[i \in E_t] + \mathbf{1}[t \ (\text{mod}) \ 2\hat{d} = 0])$
        **end if**
    **end if**
**end for**
**return FAILURE TYPE 2:** $\hat{d}$ is too small

---

fully intersecting set, that is, $d^* := \arg\min_{S \subset [m]} |S|$    subject to    $|S \cap E_h| \geq 1$   $\forall E_h \in \mathcal{E}$. With these definitions, we are ready to give our algorithmic analysis theorem.

**Theorem 1.** *Suppose $\hat{d} \geq d^*$ and $\hat{N} \geq N := |\overline{\mathcal{H}}|$. Then, with probability at least $1 - \delta$, Algorithm 1 returns successfully within $O(d^* \log(m/d^*))$ queries to the learner and the size of the returned set is $|\hat{S}| = O(\hat{d}(\log \hat{N} + \log \log m + \log(1/\delta))$.*

The proof of Theorem 1 is in Appendix D. Note that the $\log \log m$ term is negligible, for example, it can be ignored (subsumed in the constants) if $m \leq 2^N$ which is true in reasonable cases. In Section 4, we provide lower bounds showing that the number of queries to the learner is asymptotically optimal up to extraneous log factors and that the size of the returned subset is asymptotically optimal (with a computational hardness assumption) if we assume $\log \log m$ is a low-order term compared to $\log N$ and that $\delta$ is a small constant. To choose the hyperparameters $\hat{d}$ and $\xi = \Theta(\log \hat{N})$, we can use a doubling approach to find a constant factor approximation to $d^*$ and $\log N$.

### 3.3 NON-INTERPOLATING LEARNERS

For non-interpolating learners, Algorithm 1 might never return successfully even for a single training error. To remedy this, we can "squash" the training errors, that is, remove the points from $E_t$ that also appear in $S_t$. More precisely, when we calculate errors on the pool, instead of defining $E_t = \{i \in [m] : h_t(x) \neq y_i\}$, define $E_t = \{i \in [m] : h_t(x) \neq y_i \wedge i \notin S_t\}$. For the experiments, we run Algorithm 1 with this edit. This error squashing technique is used beyond our algorithm. This technique is explicitly used for the experiments in Cicalese et al. (2020) and likely a similar approach is used for Dasgupta et al. (2019) as we found experimentally that the algorithm dramatically fails, as written, without error squashing. Similar to this paper, the algorithmic analysis of Cicalese et al. (2020) assumes interpolating learners, but the practical algorithm includes error squashing.

## 4 THEORY

In this section, we first cover notation definitions, a particular class of learners, and the equivalence between machine teaching and the classic set cover problem. Next, we cover our lower bounds and results for the error squashing technique.

## 4.1 NOTATION DEFINITIONS

Let $\mathbb{Z}$ denote the set of integers and $\mathbb{Z}_+$ denote the set of positive integers. Let $[k] = \{1, 2, \ldots, k\}$. For a set $S$, let $\mathcal{P}(S)$ denote the power set of $S$. For a set $S$, let $S^k$ be the repeated Cartesian product; for example, $S^3 = S \times S \times S$. Let $\widetilde{O}(\cdot)$ denote $O(\cdot)$ where extraneous log factors are ignored. In particular, $O(f(n) \log^k(f(n))) = \widetilde{O}(f(n))$.

## 4.2 RANKED MINIMAL ERROR LEARNERS

Here, we define a type of learner that appears in the constructions for our lower bounds and is given as an example in the error squashing framework. Intuitively, a ranked minimal error learner is a learner with a hypothesis class $\mathcal{H}$ and a ranking $\sigma$ such that the learner returns the lowest ranked hypothesis that has minimal error.

**Definition 1.** *Let $\mathcal{X}$ and $\mathcal{Y}$ be finite. We say a learner $L$ is a ranked minimal error learner if there exists a (finite) hypothesis class $\mathcal{H}$ and a bijection $\sigma : \mathcal{H} \to [|\mathcal{H}|]$ such that for any $D \subset \mathcal{P}(\mathcal{X} \times \mathcal{Y})$, with $k = \min_{h \in \mathcal{H}} |\{(x, y) \in D : h(x) \neq y\}|$, $L(D) = \arg\min_{h:|\{(x,y) \in D:h(x) \neq y\}|=k} \sigma(h)$.*

Note that since we focus on fixed datasets and classification in this work, finite $\mathcal{X}$ and $\mathcal{Y}$ are not a restriction for our purposes. If we assume the ground truth labels are generated by a member of the hypothesis class, $\exists h^* \in \mathcal{H} : \forall (x, y) \in D : y = h^*(x)$, then the learner is an interpolating learner.

## 4.3 LOWER BOUNDS

We present two lower bounds that are query-based lower bounds proved using similar techniques. In particular, we construct a ranked minimal error learner with a special structure that depends on a random "key" (e.g., a random vector). We show that even if the teacher knew all the structure of the learner except the random key, any teaching algorithm will not be able to return a subset of size $\lambda d^*$ (a $\lambda$-approximation) with few queries and probability greater than $1/2$.

**Theorem 2.** *Fix any $\lambda \geq 1$ and $m \geq 2\lambda$. There exists a distribution over interpolating ranked minimal error learners and an $m$-sized dataset with optimal subset size $d^*$, such that any teaching algorithm requires $2^{\Omega(m/\lambda)}$ queries to achieve at most $\lambda d^*$ subset size with probability at least $1/2$.*

The proof is in Appendix E.1. This would appear to be very bad news: we require an exponential number of queries to even achieve a valid subset that is a factor $\lambda$ larger than the optimal subset. Fortunately, as shown later, we can not in general approximate the optimal subset to a factor of $o(\log N)$ (asymptotically strictly better than $\log N$). Furthermore, $N$ is very large in the construction for Theorem 2; so large that $\log N = \Theta(m)$. Thus, if we are content with an approximation guarantee of $O(\log N)$, then far fewer queries to the learner are required, as the existence of Algorithm 1 shows.

**Theorem 3.** *Fix any $k \geq 1$, $\ell \geq 1$, and $\lambda \geq 1$. There exists a distribution over interpolating ranked minimal error learners and a dataset of size $m = k(\lceil e\lambda \rceil)^\ell$ with optimal subset size $d^* = k$ and $N = O(k \ln(m/k) / \ln(\lambda))$, such that any teaching algorithm requires $\Omega\left(\frac{d^* \ln(m/d^*)}{\ln^2(\lambda)}\right)$ queries to achieve at most $\lambda d^*$ subset size with probability at least $1/2$.*

The proof is in Appendix E.2. Note that this implies achieving an approximation ratio of $O(\log N)$ or $O(\log m \log N)$ (as found in Cicalese et al. (2020); Dasgupta et al. (2019)) requires $\widetilde{\Omega}(d^* \ln(m/d^*))$ queries.

We now present a computational hardness lower bound on finding a subset with a small $o(\log N)$ approximation ratio to optimal, even with unlimited learner queries so that $\overline{\mathcal{H}}$ could be explicitly constructed. This result is a straightforward application of the following theorem to the set cover equivalence of machine teaching.

**Theorem 4** ((Theorem 4.4 from Feige (1998))). *If there is some $\epsilon > 0$ such that a polynomial time algorithm can approximate set cover within $(1 - \epsilon) \ln n$, then $NP \subset TIME(n^{O(\log \log n)})$.*

Let us interpret this result. For set cover instances, the "size" $n$ of a problem is measured in terms of the number of elements, our $N$. Feige (1998) defines $\mathrm{TIME}(f(n))$ to be the set of problems solvable in $f(n)$ (determnistic) time. The conclusion $\mathrm{NP} \subset \mathrm{TIME}(n^{O(\log \log n)})$ is slightly weaker

than NP $\subset$ TIME$(n^{O(1)})$ = P which would imply the famous $P = NP$. All the same, if it were to be shown that NP $\subset$ TIME$(n^{O(\log \log n)})$, it would revolutionize complexity theory, and thus, it is doubtful that such a statement can be shown to be true without significant obstacles.

Then, a direct result of set cover equivalence (Goldman & Kearns, 1995) and Theorem 4 Feige (1998) is that approximating machine teaching to within $\frac{1}{2} \ln N$ would imply NP $\subset$ TIME$(n^{O(\log \log n)})$.

### 4.4 ERROR SQUASHING FRAMEWORK

In a nutshell, our framework consists of defining a general condition for a learner, defining a teaching set definition for non-interpolating learners, then showing that achieving such a teaching set reduces, by way of error squashing, to a standard set cover problem (if the learner meets the condition). This means that the error squashing technique effortlessly converts any machine teaching algorithm for interpolating learners to a machine teaching algorithm for non-interpolating learners, which inherits the same theoretical guarantees, albeit with a larger hypothesis class.

#### 4.4.1 LEARNERS INVARIANT TO CONSISTENT ADDITIONS

In this section, we first define a general condition on a learner: if a learner trained on a dataset correctly predicts a point, then additionally including that point in the dataset does not change the the learner's predictions.

**Definition 2** (Invariant to consistent additions). *We say a learner is invariant to consistent additions if $\forall D \subset \mathcal{P}(\mathcal{X} \times \mathcal{Y})$, $x \subset \mathcal{X}$, $y \subset \mathcal{Y}$: $L(D)(x) = y \implies L(D) = L(D \cup \{(x, y)\})$.*

Intuitively, for models that minimize error, adding a point with zero loss will not change the model. In particular, we have the following proposition:

**Proposition 1.** *A ranked minimal error learner is invariant to consistent additions.*

The proof of Proposition 1 can be found in Appendix F. The invariance to consistent additions is more general than ranked minimal error learners: see an example in Appendix F.7.1.

#### 4.4.2 ERROR INCLUSIVE TEACHING SETS

Recall that error squashing with machine teaching means that when the learner is passed a subset $S_t$ and returns a hypothesis $h_t$, the teacher acts as if the hypothesis makes no errors on the subset $S_t$. Machine teaching algorithms only terminate when the returned hypothesis makes no errors on the pool, which means that all the errors of $h_t$ are squashed with respect to $S_t$. For a fixed dataset and learner, we say a set $S$ is "error inclusive" if all errors of $L(S)$ are inside $S$. Formally, for a fixed dataset $\{(x_i, y_i)\}_{i=1}^m$, define $L^{(\text{err})} : \mathcal{P}([m]) \to \mathcal{P}([m])$ as the set of indices of points where errors are made by a learner trained on a subset $S$: $L^{(\text{err})}(S) = \{i \in [m] : L(\{(x_{i'}, y_{i'}) : i' \in S\})(x_i) \neq y_i\}$.

**Definition 3.** *For a fixed dataset, we say a set $S$ is "error inclusive" if $L^{(err)}(S) \subset S$.*

Error inclusivity means that $S$ contains the points useful for generalization outside $S$, as well as the (presumably) nuisance points that are hard or impossible to classify correctly, such as incorrectly labeled ground truth. An error squashing machine teaching algorithm, if it returns successfully, must return an error inclusive set. An important consequence of error inclusivity is that, for a learner invariant to consistent additions, the learner trained on the set will yield the same result as training on the entire dataset.

**Proposition 2.** *Suppose a learner $L$ is invariant to consistent additions. If a set $S$ is error inclusive, $L^{(err)}(S) = L^{(err)}([m])$.*

The proof of Proposition 5 is in Appendix F.6. Interpreted further, error inclusivity implies that the number of errors from training on $S$ is the same as the number of errors from training on the entire dataset, $|L^{(\text{err})}(S)| = |L^{(\text{err})}([m])|$. Furthermore, error inclusivity implies $L^{(\text{err})}([m]) \subset S$, meaning that $S$ contains the errors of training on the full dataset. Error inclusivity is a condition on a set that we will use as the teaching set condition for non-interpolating learners.

Note that Cicalese et al. (2020) introduce another teaching set condition for non-interpolating learners, $k$-extended teaching sets. In Appendix F, we compare $k$-extended teaching sets and error inclusive teaching sets for ranked minimal errors. We find they are remarkably similar, though neither is stronger than the other: there exist examples where the smallest $k$-extended teaching set is of size $\Theta(m)$ but there exists an error inclusive set of size $\Theta(1)$, and vice versa. However, we show error inclusive sets are much easier to design algorithms for.

### 4.4.3 EXTENDED ERROR SETS

Recall that in the context of machine teaching with interpolating learners, we define the errors of a hypothesis as $E_h = \{i \in [m] : h(x_i) \neq y_i\}$ and let $\mathcal{E} = \{E_h : h \in \overline{\mathcal{H}}\} - \{\emptyset\}$ be the error sets. Then a set $S \subset [m]$ is a valid teaching set if $S$ fully intersects $\mathcal{E}$ (intersects each element of $\mathcal{E}$). Similarly, black-box machine teaching is like a game where elements of $\mathcal{E}$ are revealed by learner queries and a solution $S \subset [m]$ must intersect every element. In fact, there is an equivalence between these two views, observed in Goldman & Kearns (1995) as the relationship between set cover and machine teaching. For a more complete exposition, see Appendix F.

Define the *extended error sets* as $\mathcal{E}^+ = \{L^{(\text{err})}(S) \setminus S : S \subset [m]\} - \{\emptyset\}$. For interpolating models, $\mathcal{E}^+ = \mathcal{E}$. If we find a set $S \subset [m]$ that fully intersects $\mathcal{E}^+$, we are guaranteed that when the learner is trained on $S$, there will be no errors outside of $S$: $S$ is error inclusive. Not only this, we show that, in fact, fully intersecting $\mathcal{E}^+$ is equivalent to error inclusivity.

**Proposition 3.** *Fix a dataset. If a learner is invariant to consistent additions, $S$ fully intersects $\mathcal{E}^+$ if and only if $S$ is error inclusive.*

Thus, we show that (for learners invariant to consistent additions) the error squashing technique converts machine teaching with non-interpolating learners to set cover, though of an expanded set. Then, black-box machine teaching algorithms (which are online set cover algorithms in disguise) can be run with the same theoretical guarantees, though with a larger $N$. Note that the same is not true of $k$-extended teaching sets (Cicalese et al., 2020) which require solving "generalized set cover" that is harder to design algorithms for (e.g. Cicalese et al. (2020) does not implement any).

## 5 EXPERIMENTS

We evaluate 7 subset selection methods: **Random** sampling, **Entropy** sampling (Lewis & Gale, 1994), **Forgetting** events (Toneva et al., 2018), **GraNd** (Paul et al., 2021), **DHPZ** (Dasgupta et al., 2019), **CFLM** (Cicalese et al., 2020), and our introduced algorithm. Although there are many coreset selection techniques, we focused on the methods that achieve the highest accuracy regardless of computation cost. We compare to Entropy, Forgetting, and GraNd as the coreset selection methods because they perform the best in the evaluation on CIFAR10 by Guo et al. (2022) in our dataset subset size regime (20% to 60% of CIFAR10). More details can be found in Appendix B.

We compare results on 6 common image datasets (all but CINIC10[1] retrieved using `torchvision`) and use the predefined train/test splits. The six datasets are: CIFAR10 (Krizhevsky, 2009), CIFAR100 (Krizhevsky, 2009), CINIC10 (Darlow et al., 2018), FashionMNIST (FMNIST) (Xiao et al., 2017), MNIST (LeCun, 1998), and SVHN (Netzer et al., 2011).

We evaluate using three architectures: Myrtle, VGG, and ResNet10. Myrtle was created by Page (2018) by stripping away parts of ResNet and modifying the training procedure while balancing between training speed and accuracy on CIFAR-10. The end result is a network that achieves 96% on CIFAR-10 in three minutes of training time. VGG (Simonyan & Zisserman, 2015) and ResNet10 (He et al., 2016) are created according to a common pytorch library for the CIFAR datasets[2]. More details including training hyperparameters can be found in Appendix B.

### 5.1 RESULTS

We first show results (see Figure 1) for the seven methods on all six datasets with one of the architectures, ResNet10. We note that there is a variety of performances from the coreset methods though the

---

[1]CINIC10 downloaded from `https://datashare.ed.ac.uk/handle/10283/3192`
[2]`https://github.com/kuangliu/pytorch-cifar`

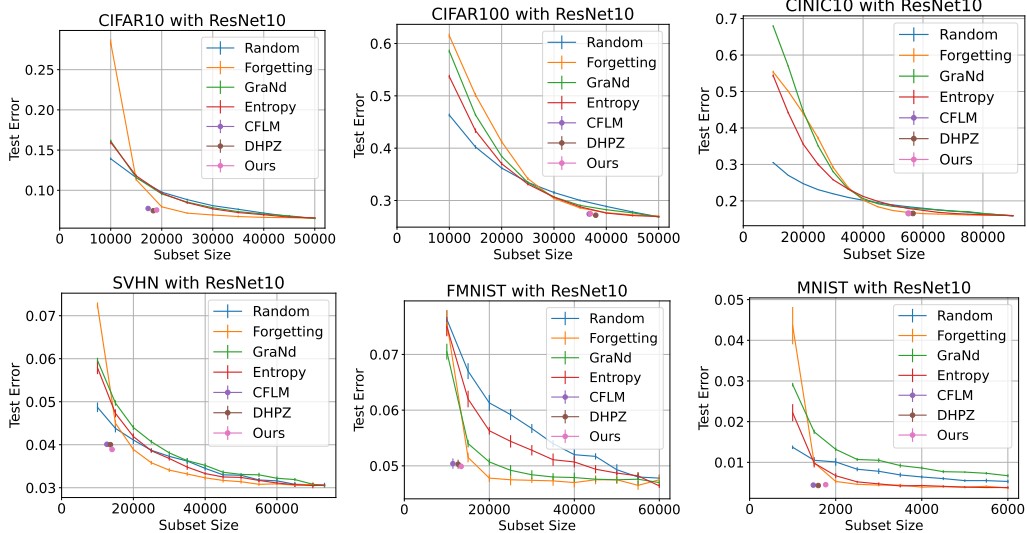

Figure 1: Plots for ResNet10 across six datasets. Error bars are from training with 10 replications on a subset.

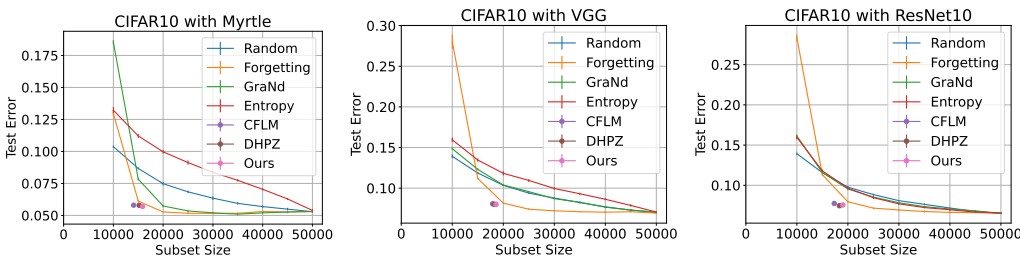

Figure 2: Plots for CIFAR10 across three model architectures. Error bars are from training with 10 replications on a subset.

forgetting events technique consistently performs the best (lower test error). The machine teaching approaches behave similarly to each other, and outperform all coreset methods.

Note that in a few cases (SVHN and FMNIST), achieving near-zero pool error is insufficient to achieve minimal test error with the data. This is surprising because the models trained on the machine teaching subsets get less than 0.01% error on the pool outside of the subset, meaning that every point is either seen or predicted correctly by the model.

Next, we study how varying the model architecture impacts performance. We show results for the seven methods with the three architectures on CIFAR10 in Figure 2. We can draw approximately the same conclusions, though surprisingly, **Entropy** performs *worse than random* on non-ResNet architectures. We note that while Entropy performs better than random in Guo et al. (2022) and Coleman et al. (2020), both these papers use ResNet architectures.

Full results for all combinations of datasets and architectures can be found in Appendix C. Although we do not measure or report wall clock time, a proxy is the number of times the model is trained. For **Entropy** and **Forgetting**, the model is trained once, for **GraNd**, the model is trained ten times, and the number of iterations for the machine teaching methods are shown in Table 1.

Finally, we perform cross-architecture experiments where model architecture used during subset selection is different from the model architecture used for evaluation. Interestingly, the behavior of the machine teaching methods differers dramatically, from transferring as well as random to transferring as well as the full dataset. See Figure 3 for an example. All 54 combinations of datasets, subset selection architectures, and evaluation architectures can be found in Appendix C.

Table 1: This table shows the number of queries to the learner for the machine teaching methods with the ResNet10 architecture. Results for the other architectures can be found in Appendix C.

| Dataset | CIFAR10 | CIFAR100 | CINIC10 | SVHN | FMNIST | MNIST |
|---|---|---|---|---|---|---|
| DHPZ | 147 | 142 | 192 | 99 | 111 | 39 |
| CFLM | 116 | 139 | 205 | 97 | 82 | 36 |
| Ours | 141 | 159 | 197 | 122 | 110 | 34 |

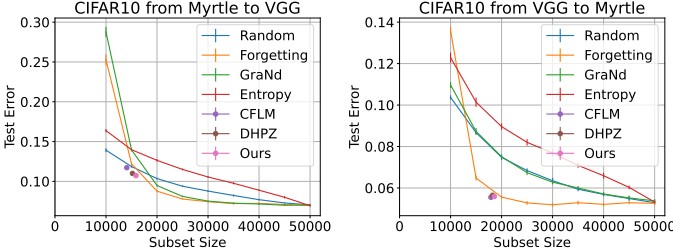

Figure 3: Example cross-architecture plots where the subset selection architecture differs from the evaluation architecture. Error bars are from training on a subset with 10 replications.

# 6 RELATED WORK

We note a number of works, sometimes known as coreset methods, focus on reducing the time to train a model once by training with less data, perhaps with a drop in performance (Coleman et al., 2020; Mirzasoleiman et al., 2020; Paul et al., 2021; Killamsetty et al., 2021b;a). A key challenge in that formulation is creating a data selection algorithm that is computationally fast enough to avoid negating the computational gain of training on less data. In this work, we focus on data selection when we will train multiple times on the data subset (curriculum learning, continual learning, etc), so computational cost is less of an issue.

Data subset selection without knowing the labels of the datapoints before a point is selected is known as active learning (Settles, 2009), a strictly harder problem. Note that an optimal data subset collected with full knowledge of the labels is an upper bound on how well an active learning algorithm can perform.

As mentioned throughout this work, machine teaching (Goldman & Kearns, 1995; Zhu et al., 2018) is closely related to subset selection; especially Dasgupta et al. (2019) and Cicalese et al. (2020). In these works, the specification for machine teaching with black-box models is a bit different from our formulation. In particular, those works require the teaching set to be built up iteratively so that the queried subsets form a nested sequence. While natural from a machine teaching perspective, this requirement is unnecessary for subset selection as we can train on arbitrary subsets before returning an arbitrary subset. This less restrictive specification is likely why the $\log m$ factor can be trimmed from our algorithm subset size; see Alon et al. (2009) for more details on a related lower bound.

An adjacent problem setting is that of "Dataset Distillation" (Wang et al., 2018; Nguyen et al., 2021) or "Dataset Condensation" (Zhao et al., 2020; Zhao & Bilen, 2021) where a very small (e.g. 100 images) dataset of *synthetic* images yields decent, but markedly sub-par, performance. Additionally, the optimized synthetic images can be very far from the natural data distribution.

# 7 DISCUSSION

In this work, we bring together the neural network coreset algorithm and recent machine teaching literature in data subset selection. At the same time, we streamline the theory for machine teaching by proving the correctness of an effective heuristic used in machine teaching implementations and by closing theoretical gaps in the asymptotic number of learner queries and size of the returned subset. More broadly, we hope the insights produced in this work will spur further research in selecting informative datasets and the role of data in learning.

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

Table 2: This table shows the value of the parameters that are related to a $\log N$ estimate. For example, $\lambda$ for DHPZ, $\xi$ for ours, and the number of sampling repetitions for CFLM. All parameters are the minimal element from the set $\{2^k 10^3 : k \in \mathbb{Z}\}$ such that the algorithm returns successfully.

| Dataset | $\lambda$ (DHPZ) | Sampling Repeats (CFLM) | $\xi$ (Ours) |
|---|---|---|---|
| CIFAR10 | 1000 | 2000 | 1000 |
| CIFAR100 | 8000 | 16000 | 4000 |
| CINIC10 | 8000 | 16000 | 8000 |
| SVHN | 1000 | 2000 | 500 |
| FMNIST | 1000 | 2000 | 500 |
| MNIST | 125 | 250 | 125 |

## A  APPENDIX OVERVIEW

This appendix is split into 5 sections:

- Additional Experimental Results
- Experimental Details
- Upper Bound Proofs
- Lower Bound Proofs
- Error Squashing Framework Exposition and Proofs

### A.1  NOTATION DEFINITIONS

Let $\mathbb{Z}$ denote the set of integers and $\mathbb{Z}_+$ denote the set of positive integers. Let $[k] = \{1, 2, \ldots, k\}$. For a set $S$, let $\mathcal{P}(S)$ denote the power set of $S$. For a set $S$, let $S^k$ be the repeated Cartesian product; for example, $S^3 = S \times S \times S$.

Let $\widetilde{O}(\cdot)$ denote $O(\cdot)$ where extraneous log factors are ignored. For example, if an algorithm has a $O(n \log n)$ runtime, it also has a $\widetilde{O}(n)$ runtime. Note that we cannot ignore all log terms (otherwise we can drop everything: $O(n) = O(\log(\exp(n)))$, only log factors where the non-log version of an expression also appears.

## B  EXPERIMENTAL DETAILS

### B.1  MACHINE TEACHING IMPLEMENTATION DETAILS

The machine teaching techniques, including ours, have three implementation adaptations. First, all methods are run with error squashing. Second, instead of terminating when no pool errors (outside of the training errors) are made, the algorithms return when the number of pool errors is less than $10^{-4} m$. Third, to avoid training on very small subsets where training instabilities occur, when a machine teaching algorithm attempts to query a subset with fewer than $10^4$ points, we randomly sample enough points so that the total number of points is exactly $10^4$. Because MNIST requires significantly fewer points, we randomly sampled to $10^3$ points.

All machine teaching algorithms include a hyperparameter that roughly corresponds to $\log N$. For each method and dataset, we tuned over the set $\{2^k 10^3 : k \in \mathbb{Z}\}$ to find the minimal hyperparameter value where the algorithm returned successfully. See Table 2. Our proposed algorithm additionally has the hyperparameter $\hat{d}$ which was set to $10^3$ for all experiments except for MNIST it was set to $10^2$.

### B.2  CORESET IMPLEMENTATION DETAILS

Forgetting scores were calculated while training the subset selection model once. Ten different subset selection models were trained for the ten different runs.

Table 3: This table shows the number of queries to the learner for the machine teaching methods with the Myrtle architecture.

| Dataset | DHPZ | CFLM | Ours |
|---------|------|------|------|
| CIFAR10 | 128 | 108 | 125 |
| CIFAR100 | 126 | 150 | 146 |
| CINIC10 | 144 | 197 | 201 |
| SVHN | 76 | 89 | 87 |
| FMNIST | 105 | 73 | 102 |
| MNIST | 47 | 39 | 42 |

Entropy was calculated using the softmax probability distribution after training until completion. Ten different subset selection models were trained for the ten different runs.

GraNd was calculated as the L2 norm of the gradient of cross entropy loss with respect to the last layer's weights and biases, averaged over ten runs. All ten replications used the same GraNd scores (averaged from ten runs).

### B.3 MYRTLE DETAILS

Details for the Myrtle architecture can be found here `https://github.com/davidcpage/cifar10-fast`. Specifically, we use the updated version that uses a preprocessing whitening block and a weighted KL and cross entropy loss, as well as tricks such as Ghost BatchNorm, two sets of SGD optimizers, and exponential moving averages. Note that we use neither CutOut augmentation nor test time augmentation, as we stick to training augmentation with crops and flips. We keep most hyperparameters the same as the default ones, except we train for $50 * 50000$ steps and set the base learning rate to $0.4$.

### B.4 VGG AND RESNET10 DETAILS

The code for the VGG and ResNet10 can be found at `https://github.com/kuangliu/pytorch-cifar`. VGG was created with the command `VGG('VGG13')` and ResNet10 was created with the command `ResNet(BasicBlock, [1, 1, 1, 1])`. Both architectures were trained using SGD with momentum of $0.9$ and weight decay $0.0005$, and a triangular learning rate schedule with a maximum learning rate of $0.1$. Cross entropy loss and a gradient scaler were also used. The number of gradient steps was set to be $50 \times m$ (i.e. 50 epochs for the full training, more for subsets).

## C ADDITIONAL EXPERIMENTS

We first present the number of iterations for the machine teaching methods with Myrtle and VGG in Table 3 and Table 4.

This appendix includes the full experimental plots. In the first figure (Figure 4), all three architectures are run with all six datasets. The following three figures are cross-architecture experiments and are grouped by the evaluation architecture. See Figure 5 for Myrtle evaluation, Figure 6 for VGG evaluation, and Figure 7 for ResNet10 evaluation. "CIFAR10 with Myrtle" means Myrtle was used for both subset selection and evaluation, while "CIFAR10 from VGG to ResNet10" means VGG was used for subset selection and ResNet10 was used for evaluation.

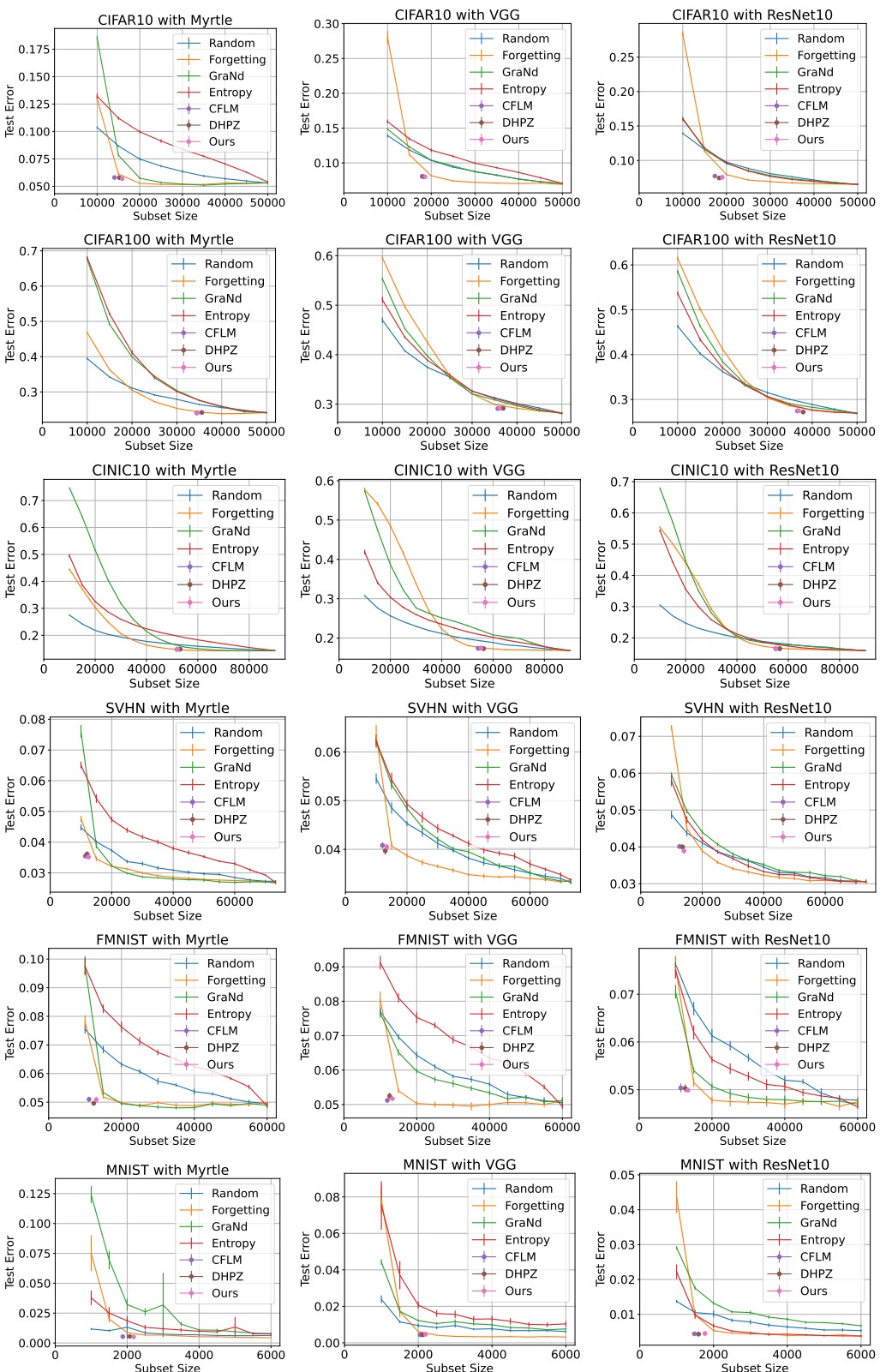

Figure 4: Plots for three architectures by six datasets.

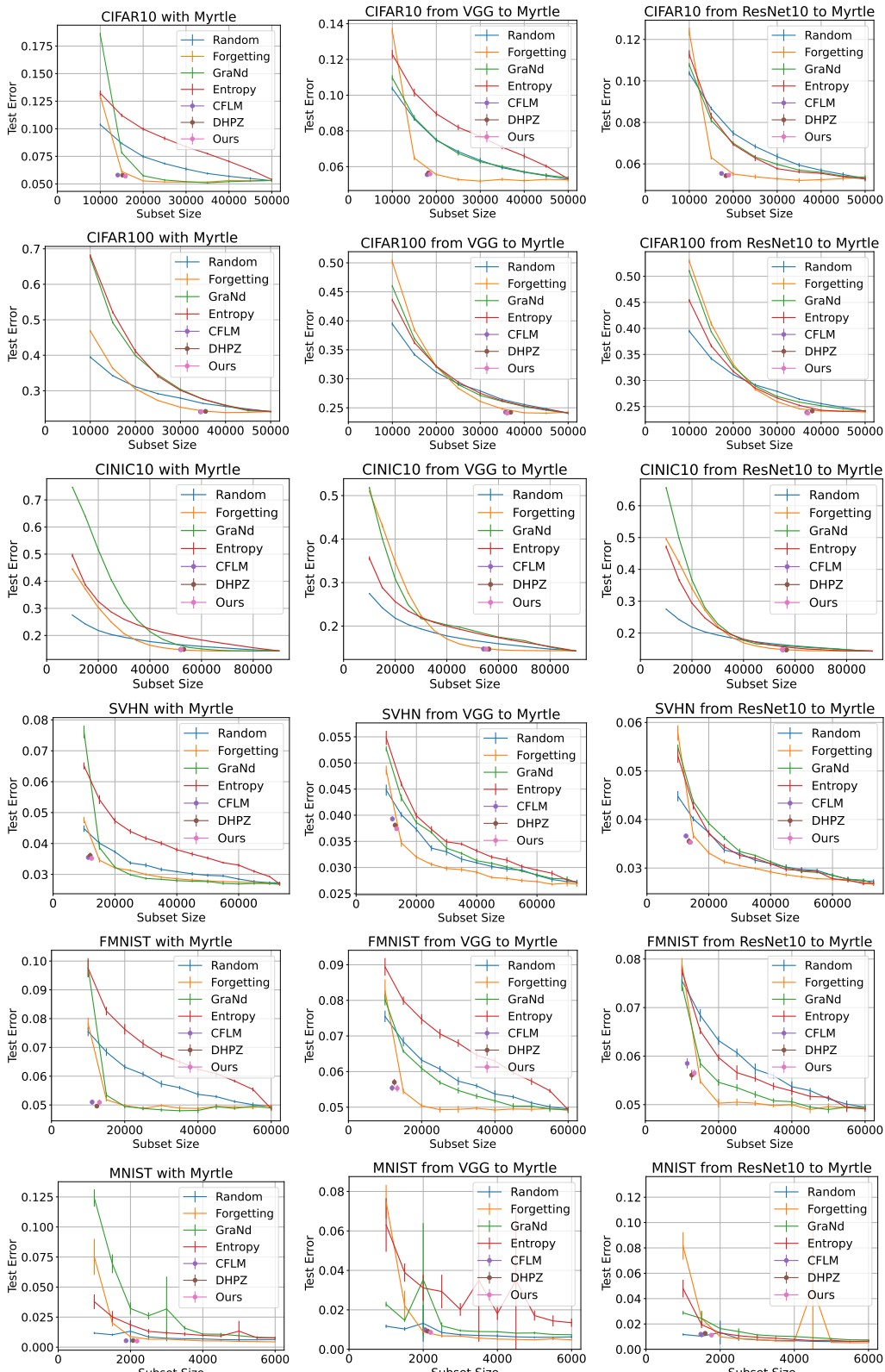

Figure 5: Cross-architecture evaluation on Myrtle.

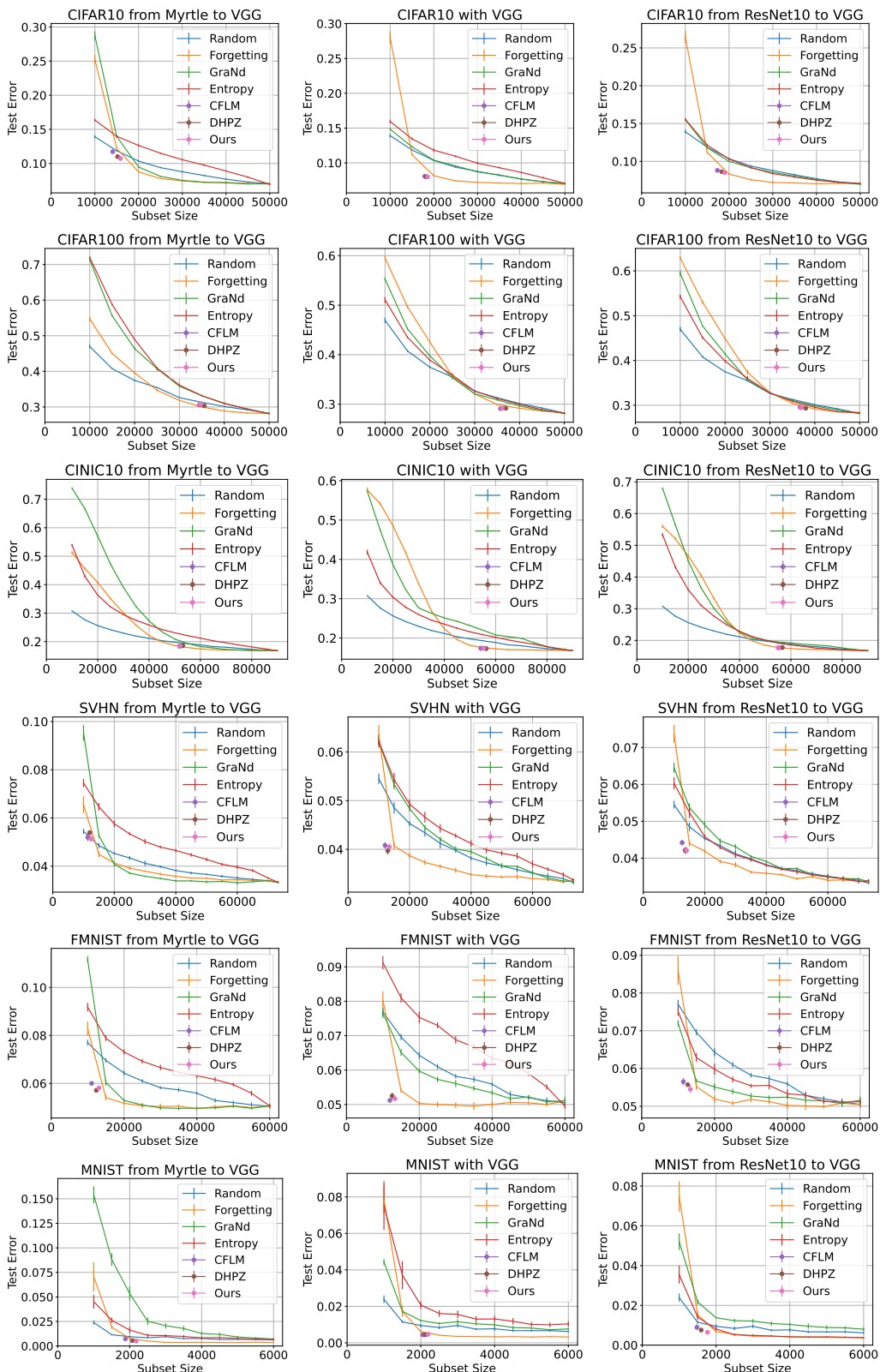

Figure 6: Cross-architecture evaluation on VGG.

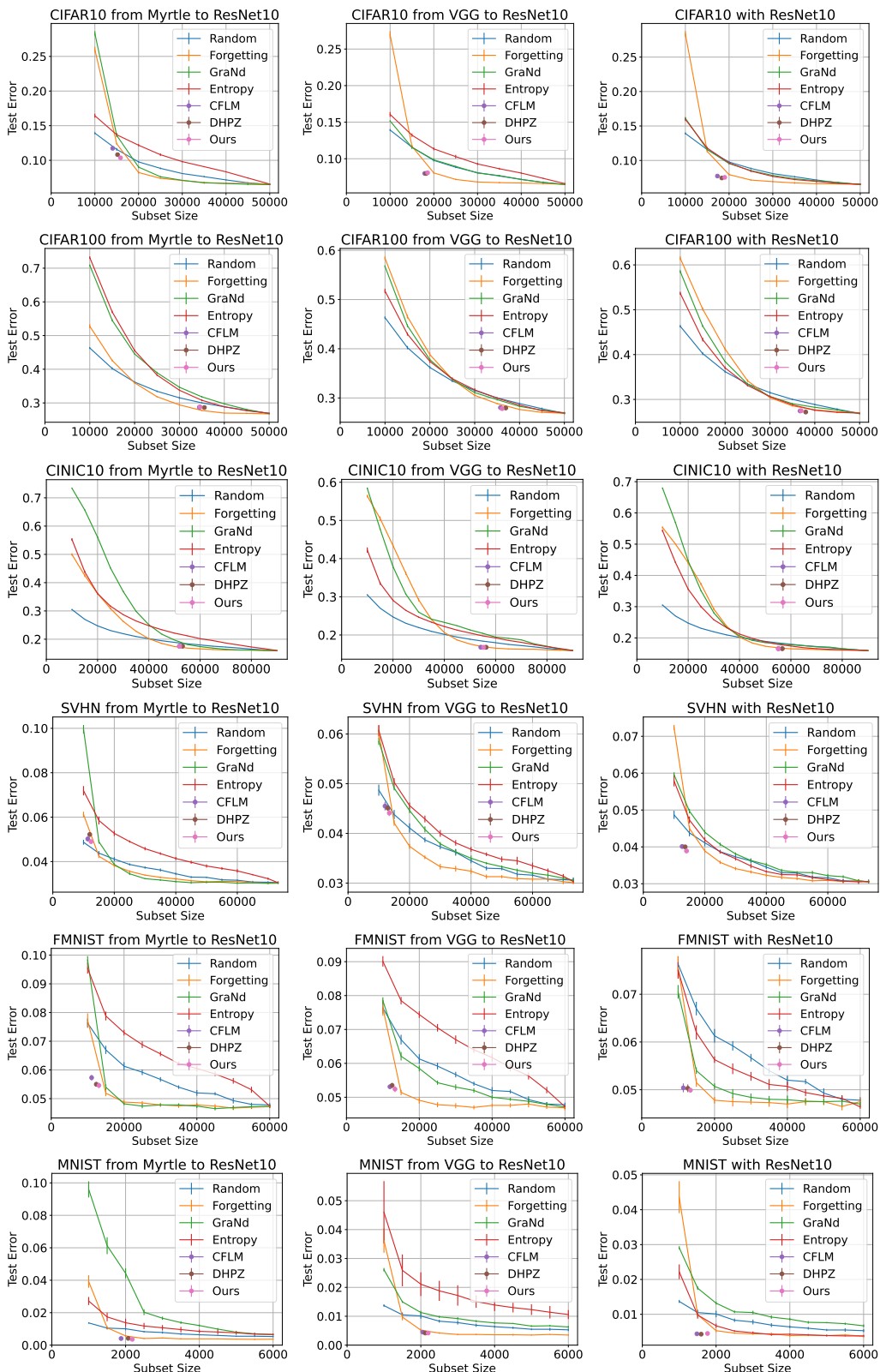

Figure 7: Cross-architecture evaluation on ResNet10.

Table 4: This table shows the number of queries to the learner for the machine teaching methods with the VGG architecture.

| Dataset | DHPZ | CFLM | Ours |
|---------|------|------|------|
| CIFAR10 | 143 | 158 | 116 |
| CIFAR100 | 123 | 123 | 143 |
| CINIC10 | 202 | 158 | 195 |
| SVHN | 90 | 79 | 113 |
| FMNIST | 87 | 99 | 96 |
| MNIST | 42 | 45 | 38 |

## D  UPPER BOUND PROOFS

In this section, we prove the following theorem through a sequence of lemmas.

**Theorem 1.** *Suppose $\hat{d} \geq d^*$ and $\hat{N} \geq N := |\overline{\mathcal{H}}|$. Then, with probability at least $1 - \delta$, Algorithm 1 returns successfully within $O(d^* \log(m/d^*))$ queries to the learner and the size of the returned set is $|\hat{S}| = O(\hat{d}(\log \hat{N} + \log \log m + \log(1/\delta))$.*

Throughout this section, we implicitly assume $\hat{d} \geq d^*$ and $\hat{N} \geq N$.

### D.1  HIGH PROBABILITY EVENT

Set $\overline{T} = 2\hat{N}\lfloor \log_2 m \rfloor$ and recall that $\xi = \ln(4\hat{N}^2 \log_2(m)/\delta)$

$$G_{h,t} = \left\{ |S_t \cap E_h| \geq 1 \vee \sum_{i \in E_h} p_{t,i} < \xi \right\} \tag{1}$$

$$G = \bigcap_{t=1}^{\overline{T}} \bigcap_{h \in \overline{\mathcal{H}}} G_{h,t} \tag{2}$$

**Lemma 1.** *Event $G$ holds with probability $1 - \frac{\delta}{2}$.*

*Proof.* Fix any $h \in \overline{\mathcal{H}}$ and $t \leq \overline{T}$.

Note $\neg G_{h,t} = \left\{ |S_t \cap E_h| = 0 \wedge \sum_{i \in E_h} p_{t,i} \geq \xi \right\}$

Examine the random variable $|S_t \cap E_h| = \sum_{i \in E_h} \text{Bernoulli}(p_{t,i})$. Set $P_{h,t} = \sum_{i \in E_h} p_{t,i}$. By a Chernoff bound and taking the limit $\delta \to 1$,

$$\Pr(|S_t \cap E_t| < (1-\delta)P_{h,t}) \leq \left( \frac{e^{-\delta}}{(1-\delta)^{(1-\delta)}} \right)^{P_{h,t}} \tag{3}$$

$$\Pr(|S_t \cap E_h| = 0) \leq \exp(-P_{h,t}) \tag{4}$$

Therefore,

$$\Pr(\neg G_{h,t}) \leq \exp(-\xi) \tag{5}$$

Then, by a union bound,

$$\Pr(\neg G) = \Pr\left(\bigcup_{t=1}^{\overline{T}} \bigcup_{h \in \overline{\mathcal{H}}} \neg G_{h,t}\right) \tag{6}$$

$$\leq \sum_{t=1}^{\overline{T}} \sum_{h \in \overline{\mathcal{H}}} \Pr(\neg G_{h,t}) \tag{7}$$

$$\leq \overline{T} N \exp(-\xi) \tag{8}$$

$$\leq 2\hat{N} \log_2(m) N \frac{\delta}{4\hat{N}^2 \log_2(m)} \tag{9}$$

$$\leq \frac{\delta}{2} \tag{10}$$

$\square$

**Lemma 2.** *If $G$ holds, then for $t \leq \overline{T}$, $\sum_{i \in E_t} p_{t,i} < \xi$.*

*Proof.* By definition of the interpolating model, $|S_t \cap E_t| = 0$, therefore, since $G$ holds, $\sum_{i \in E_t} p_{t,i} < \xi$. $\square$

Note this implies that if $G$ holds and the algorithm terminates before $\overline{T}$, we won't return **FAILURE TYPE 1**

### D.2 BOUND ON NUMBER OF ITERATIONS

**Lemma 3.** *If $G$ holds and $\hat{N} < 2d^*$, then the algorithm returns successfully within $\hat{N} \leq 2d^* k_0$ iterations.*

*Proof.* First note that $\overline{T} = 2\hat{N} \log_2(m) \geq \hat{N}$.

Note that, under $G$ and if the probabilities are not halved, a hypothesis cannot be returned twice by the learner. This is because when a learner returns $h_t$, for the next iterations, the weight will be at least $\xi$ and under $G$, by Lemma 2, $h_t$ cannot be returned again. Thus, if $N \leq \hat{N} < 2\hat{d}$, the learner will return all hypotheses before halving. $\square$

**Lemma 4.** *If $G$ holds and $\hat{N} \geq 2d^*$, Algorithm 1 returns successfully within $2d^* k_0 + 1$ iterations.*

*Proof.* Fix a subset solution $S^*$ that has the optimal size $d^*$.

Let $T = 2d^* k_0 + 1$.

Note that $T \leq 2\hat{d}\lfloor \log_2(m) \rfloor \leq \hat{N}\lfloor \log_2(m) \rfloor \leq \overline{T}$.

Suppose we have not returned by iteration $T$; then, by assumption $\forall t \leq T : |E_t| \geq 1$.

Note that $E_t \in \mathcal{E}$, so $|S^* \cap E_t| \geq 1$.

$$\sum_{i \in S^*} \mathbf{1}[i \in E_t] \geq 1 \tag{11}$$

$$\sum_{i \in S^*} \sum_{t=1}^{T} \mathbf{1}[i \in E_t] \geq T \tag{12}$$

Note that at every iteration, $k_{t,i}$ for $i$ in $E_t$ decrease by at least one. By $G$ holding, $E_t \leq \xi$. Thus, $\Delta_t \geq 1$. And further, if $k_{t,i} = 0$, then $p_{t,i} = 1$, $i \in S_t$, and $i \notin E_t$.

$$k_{T+1,i} \leq k_0 - \sum_{t=1}^{T} \mathbf{1}[i \in E_t] + \sum_{t=1}^{T} \mathbf{1}[t \ (\text{mod}) \ 2\hat{d} = 0] \tag{13}$$

But since $k_{T+1,i} \geq 0$,

$$k_0 - \sum_{t=1}^{T} \mathbf{1}[i \in E_t] + \frac{T}{2\hat{d}} \geq 0 \tag{14}$$

$$d^* k_0 - \sum_{i \in S^*} \sum_{t=1}^{T} \mathbf{1}[i \in E_t] + \frac{d^* T}{2\hat{d}} \geq 0 \tag{15}$$

$$d^* k_0 - T + \frac{d^* T}{2\hat{d}} \geq 0 \tag{16}$$

$$d^* k_0 \geq \left(1 - \frac{d^*}{2\hat{d}}\right) T \tag{17}$$

$$d^* k_0 \geq \frac{1}{2} T \tag{18}$$

$$2d^* k_0 \geq T \tag{19}$$

which is a contradiction. So the algorithm must have returned before $T$.

$\square$

Thus, in either case ($\hat{N} < 2\hat{d}$ or $\hat{N} \geq 2\hat{d}$), the algorithm terminates before $\overline{T}$ iterations and within $2d^* k_0 + 1$ iterations.

### D.2.1 SIZE OF RETURNED SUBSET

We now analyze the size of the returned set $|\hat{S}|$. First, note that at every iteration, $\mathbb{E}[|S_t|] = \sum_i p_{t,i}$. We can think of bounding the size of $|\hat{S}|$ as bounding the size of each $|S_t|$ which involves bounding the mean ($\sum_i p_{t,i}$) and the deviation from the mean. First we bound the deviation.

**Lemma 5.** *If $\sum_{i=1}^{m} p_{t,i} \leq L$ for all $t \leq \overline{T}$, then, with $1 - \delta$ probability, $|S_t| \leq \max(e^2 L, \ln(2\overline{T}/\delta)$ for each $t \leq \overline{T}$ with probability at least $1 - \frac{\delta}{2}$.*

*Proof.* Note that $|S_t| = \sum_{i=1}^{m} \text{Bernoulli}(p_{t,i})$.

Define $P_t = \sum_{i=1}^{m} p_{t,i}$. Note $0 \leq P_t \leq L$.

Fix $t$. By a Chernoff bound, for any $\kappa > 0$,

$$\Pr[|S_t| \geq (1+\kappa)P_t] \leq \left(\frac{e^\kappa}{(1+\kappa)^{(1+\kappa)}}\right)^{P_t} \tag{20}$$

$$\tag{21}$$

Define $c = \max(e^2, \ln(2\hat{N}\overline{T})/L)$.

Let $(1+\kappa)P_t = cL$, so $\kappa = \frac{cL}{P_t} - 1$.

$$\Pr[|S_t| \ge cL] \le \frac{e^{cL-P_t}}{(cL/P_t)^{(cL)}} \tag{22}$$

$$\le \frac{e^{cL}}{(c)^{(cL)}} \tag{23}$$

$$= \left(\frac{e}{c}\right)^{cL} \tag{24}$$

$$\le e^{-cL} \tag{25}$$

$$\Pr[|S_t| \ge \max(e^2 L, \ln(2\overline{T}/\delta))] \le \frac{\delta}{2\overline{\overline{T}}} \tag{26}$$

Finally, a union bound over all iterations $t \le \overline{T}$ yields the result. $\square$

**Lemma 6.** *If event $G$ holds, for $t \le \overline{T}$,*

$$\sum_{i=1}^{n} p_{t,i} \le 8\hat{d}\xi \tag{27}$$

*Proof.* We will show something slightly stronger, for $t \le \overline{T}$,

$$\sum_{i=1}^{n} p_{t,i} \le (2\hat{d} + ((t-1) \ (\mathrm{mod}) \ (2\hat{d})))(2\xi) \tag{28}$$

We prove this by induction. As a base case, for $t = 1$,

$$\sum_{i=1}^{n} p_{1,i} = m2^{-k_0} \tag{29}$$

$$\le m2^{\log_2(m/\hat{d})+1} \tag{30}$$

$$= 2\hat{d} \tag{31}$$

$$\le (2\hat{d} + 0)(2\xi) \tag{32}$$

For the inductive case, we examine two cases: $t \ (\mathrm{mod}) \ (2\hat{d}) \ne 0$ and $t \ (\mathrm{mod}) \ (2\hat{d}) = 0$. In both cases, we rely on Lemma 2 and event $G$ holding. For the first case,

$$\sum_{i=1}^{n} p_{t+1,i} = \sum_{i=1}^{n} 2^{-k_{t+1,i}} \tag{33}$$

$$\leq \sum_{i=1}^{n} 2^{-k_{t,i}+\Delta_t \mathbf{1}[i \in E_t]} \tag{34}$$

$$\leq \sum_{i=1}^{n} 2^{-k_{t,i}} + \sum_{i \in E_t} 2^{-k_{t,i}+\Delta_t} \tag{35}$$

$$= \sum_{i=1}^{n} p_{t,i} + 2^{\Delta_t} \sum_{i \in E_t} p_{t,i} \tag{36}$$

$$\leq \sum_{i=1}^{n} p_{t,i} + 2 \frac{\xi}{\sum_{i \in E_t} p_{t,i}} \sum_{i \in E_t} p_{t,i} \tag{37}$$

$$= \sum_{i=1}^{n} p_{t,i} + 2\xi \tag{38}$$

$$\leq (2\hat{d} + ((t-1) \ (\mathrm{mod}) \ (2\hat{d}))2\xi + 2\xi \tag{39}$$

$$\leq (2\hat{d} + (((t+1)-1) \ (\mathrm{mod}) \ r))2\xi \tag{40}$$

For the second case,

$$\sum_{i=1}^{n} p_{t+1,i} = \sum_{i=1}^{n} 2^{-k_{t+1,i}} \tag{41}$$

$$\leq \sum_{i=1}^{n} 2^{-k_{t,i}+\Delta_t \mathbf{1}[i \in E_t]-1} \tag{42}$$

$$\leq \frac{1}{2}\left[\sum_{i=1}^{n} 2^{-k_{t,i}} + \sum_{i \in E_t} 2^{-k_{t,i}+\Delta_t}\right] \tag{43}$$

$$\leq \frac{1}{2}\left[\sum_{i=1}^{n} p_{t,i} + 2\xi\right] \tag{44}$$

$$= \frac{1}{2}\left((2\hat{d} + ((2\hat{d}-1)+1)2\xi\right) \tag{45}$$

$$= 4\hat{d}\xi \tag{46}$$

$$= \left(2\hat{d} + ((t+1)-1) \ (\mathrm{mod}) \ (2\hat{d})\right)2\xi \tag{47}$$

$$\square$$

Putting the bound on the mean and deviation of the mean, we achieve a bound on $|S_t|$ and thus $|\hat{S}|$.

**Proposition 4.** *With probability* $1 - \delta$,

$$|\hat{S}| \leq \max(e^2 8\hat{d}\xi, \ln(2\overline{T}/\delta)) \tag{48}$$

$$\leq e^2 8\hat{d}\xi \tag{49}$$

$$= O\left(\hat{d}(\log(\hat{N}) + \log\log m + \log(1/\delta))\right) \tag{50}$$

## E    LOWER BOUNDS

In this section, we prove three lower bounds.

### E.1 PROOF OF QUERY LOWER BOUND FOR LARGE N

**Theorem 2.** *Fix any $\lambda \geq 1$ and $m \geq 2\lambda$. There exists a distribution over interpolating ranked minimal error learners and an $m$-sized dataset with optimal subset size $d^*$, such that any teaching algorithm requires $2^{\Omega(m/\lambda)}$ queries to achieve at most $\lambda d^*$ subset size with probability at least $1/2$.*

*Proof.* Let the dataset include $m$ points that are all labeled 0.

Let $K \subset [m]$ be randomly chosen such that $K \subset [m]$ and $|K| = \lfloor m/(2\lambda) \rfloor \geq 1$. $K$ is the random "key".

$$L(\{\vec{D}_i : i \in S\})(x_j) = \begin{cases} 1 & K \not\subset S, |S| \leq m/2, j \notin S \\ 0 & \text{otherwise} \end{cases} \tag{51}$$

We show that the learner can be written as a ranked minimal error learner.

Define

$$h_S(x_j) = \mathbf{1}[j \notin S] \tag{52}$$
$$\mathcal{H} = \{h_S : S \subset [m] : K \not\subset S \wedge |S| \leq m/2\} \cup \{h_{[m]}\} \tag{53}$$

and let $R(h_S) < R(h_{S'})$ if $|S| < |S'|$ (i.e. the learner prefers $h_S$ with smaller $S$).

Note that this learner is an interpolating learner because $L(K)$ has zero errors. Furthermore, $d^* = |K| = \lfloor m/(2\lambda) \rfloor$.

Note that any set of size $|S| = m/2 + 1$ achieves no errors (error inclusive) but the size is more than $d^*\lambda$.

Furthermore, to achieve $d^*\lambda$, a necessary condition for any algorithm is to query a subset where $|S| \leq m/2 \wedge K \subset S$. Otherwise, there will be many errors ($\geq m/2$) and no information will be gained on $K$. This is not sufficient, but is necessary.

Furthermore, for any subset $|S| \leq m/2$, the probability that $K \subset S$ is at most $1/2^{|K|}$.

Thus, by a union bound, in $2^{|K|-1}$ iterations, there is a $1/2$ chance that the algorithm has not met the necessary condition and thus is not done. Thus, the expected number of iterations is at least

$$2^{|K|-1} = 2^{\lfloor m/(2\lambda) \rfloor - 1} = 2^{\Omega(m/\lambda)} \tag{54}$$

$\square$

### E.2 QUERY LOWER BOUND FOR SMALL N

Just for this section, define $[n] = \{0, 1, \ldots, n-1\}$. In other words, we start at zero rather than one.

Here we describe a ranked minimal error learner that requires $\widetilde{\Omega}(d^* \log(m/d^*))$ iterations (queries) to even achieve even a rough approximation of the optimal subset.

Fix any $C, \ell, k \in \mathbb{Z}_+$. Let there be a randomly chosen vector $\vec{c} \in [C]^{k\ell}$. Index $\vec{c}$ by $[k\ell]$.

For simplicity let the ground truth labels all be 0.

Let $x \in [k] \times [C]^\ell$.

Let there be $k\ell + 1$ hypotheses where one hypothesis is a hypothesis that makes no mistakes $h^* = h_\infty$. For the other $k\ell$ hypotheses, index the hypotheses by $[k\ell]$.

$$h_i(x) = \begin{cases} 1 & i//\ell = x_1 \wedge x_{2,i\%\ell} = \vec{c}_i \\ 0 & \text{otherwise} \end{cases} \tag{55}$$

where $//$ signifies integer division and $\%$ signifies modulo.

Let $R(h_i) = i$ and $R(h_\infty) = k\ell$. So $h_0$ is preferred, and we must rule out all but the last hypothesis to achieve zero error.

We assume the teacher knows everything about the structure of the hypotheses except for the randomness in defining $\vec{c}$ (the random "key").

Note that we can cover all non-optimal hypotheses with $k$ points, so $d^* = k$.

We now describe the dynamics of making queries. At every iteration $t$, the teacher can keep track of an uncertainty set for each element of $\vec{c}$. In particular, let $V_{t,i} \in \mathcal{P}([C])$ be an uncertainty set such that we know $\vec{c}_i \in V_{t,i}$ at the $t^{th}$ iteration. In particular, each uncertainty set begins as $[C]$. If the $i^{th}$ hypothesis is returned by the learner, the teacher knows $V_{t,i} = \{\vec{c}_i\}$. If we know that we covered $h_i$ with a query set $X_t$ (by knowing the ranked order of hypotheses), then we know that $\vec{c}_i \in \bigcup_{x \in X_t : x_1 = i//\ell} x_{2, i\%\ell}$.

Given the above, we can refine the problem to the following problem with states $Z_t \in [C]^{k\ell}$, actions $A_t \in [C]^{k\ell}$, and randomness $R_t \in [k\ell] \cup \{\infty\}$.

Let $Z_{t,i} = |V_{t,i}|$ be the size of the uncertainty set.

Let

$$A_{t,i} = \left| \left( \bigcup_{x \in X_t : x_1 = i//\ell} x_{2, i\%\ell} \right) \cap V_{t,i} \right| \tag{56}$$

be the number of elements within the certainty set that we cover with $X_t$.

Finally, let $R_t$ be the hypothesis returned from the learner to the teacher at the $t^{th}$ iteration.

Given a state $Z_t$, the teacher chooses $X_t$ which can be converted to an $A_t$ such that $0 \leq A_{t,i} \leq Z_{t,i}$. For any query set $X_t$ that the teacher queries, we convert this to an $A_t$ where $|X_t| = \sum_{i_1 \in [k]} \max_{i_2 \in [\ell]} A_{t, i_1 \ell + i_2}$.

The learner gives the teacher a hypothesis (and thus $R_t$) that is random with respect to the randomness in $\vec{c}$ such that

$$\Pr(R_t = i) = \prod_{j=1}^{i-1} \frac{A_{t,j}}{Z_{t,j}} \left( 1 - \frac{A_{t,i}}{Z_{t,i}} \right) \tag{57}$$

$$\Pr(R_t = \infty) = \prod_{j=1}^{k\ell} \frac{A_{t,j}}{Z_{t,j}} \tag{58}$$

In other words, $h_i$ is returned by the learner if the query $X_t$ covers the first $i-1$ hypotheses, but fails to cover the $i^{th}$ hypothesis. Finally, we return $h^* = h_\infty$ if the rest of the hypotheses are covered.

Then, on a state $Z_t$, action $A_t$, and learner's hypothesis $R_t$, we can generate the next state as

$$Z_{t+1,i} = \begin{cases} A_{t,i} & i < R_t \\ 1 & i = R_t \\ Z_{t,i} & i > R_t \end{cases} \tag{59}$$

### E.2.1 ANALYSIS FOR LOWER BOUND

The analysis hinges on examining the potential function:

$$P(Z_t) = \sum_{i=1}^{k\ell} \ln\left(\frac{C}{Z_{t,i}}\right) \tag{60}$$

First, note that the potential function is monotone increasing (not strictly) with the number of iterations. Further note that $P(Z_1) = 0$ since $Z_{1,i} = C$.

Note that at any iteration $t$ and state $Z_t$, the teacher can cover all possible (with respect to the uncertainty sets) hypotheses with exactly

$$Q(Z_t) = \sum_{i_1 \in [k]} \max_{i_2 \in [\ell]} Z_{t, i_1\ell+i_2} \tag{61}$$

points. Furthermore, all uncertainty sets cannot be covered with fewer points. For an algorithm to achieve a final subset of size of at most $\lambda k$, it must be the case that $Q(Z_t) \le \lambda k$.

**Lemma 7.** *For an algorithm to achieve a final subset at iteration $t$ of size at most $\lambda k$, it must be the case that*

$$P(Z_t) \ge k\ell \ln\left(\frac{C}{\lambda}\right) \tag{62}$$

*Proof.* This follows from solving the relaxed optimization problem where $Z_{t,i} \in \mathbb{R}$ and $1 \le Z_{t,i} \le C$.

$$\min P(Z_t) \text{ such that } Q(Z_t) \le \lambda k \tag{63}$$

which has a global solution where all elements of $Z_t$ are equal: $Z_{t,i} = \lambda$ $\qquad\square$

**Lemma 8.**
$$\mathbb{E}[P(Z_{t+1}) - P(Z_t)] \le 1 + \ln(C) \tag{64}$$

*Proof.*

$$\mathbb{E}[P(Z_{t+1}) - P(Z_t)] = \mathbb{E}\left[\sum_i \ln\left(\frac{Z_{t,i}}{Z_{t+1,i}}\right)\right] \tag{65}$$

$$= \sum_{r=0}^{\infty} \Pr(R_t = r)\left[\sum_{i=0}^{r-1} \ln\left(\frac{Z_{t,i}}{A_{t,i}}\right) + \ln\left(\frac{Z_{t,i}}{1}\right)\right] \tag{66}$$

$$\le \ln(C) + \sum_{r=0}^{\infty} \Pr(R_t = r)\sum_{i=0}^{r-1} \ln\left(\frac{Z_{t,i}}{A_{t,i}}\right) \tag{67}$$

$$= \ln(C) + \sum_{i=0}^{k\ell-1}\sum_{r=i+1}^{\infty} \Pr(R_t = r)\ln\left(\frac{Z_{t,i}}{A_{t,i}}\right) \tag{68}$$

$$= \ln(C) + \sum_{i=0}^{k\ell-1} \Pr(R_t > i)\ln\left(\frac{Z_{t,i}}{A_{t,i}}\right) \tag{69}$$

$$= \ln(C) + \sum_{i=0}^{k\ell-1}\left(\prod_{j=0}^{i}\frac{A_{t,j}}{Z_{t,j}}\right)\ln\left(\frac{Z_{t,i}}{A_{t,i}}\right) \tag{70}$$

Define $\pi \in \mathbb{R}^{k\ell}$ where $\pi_i = \frac{A_{t,i}}{Z_{t,i}}$ so $0 \le \pi_i \le 1$. We can solve the relaxed problem of maximizing over $\pi$ to upper bound the expression.

$$\mathbb{E}[P(Z_{t+1}) - P(Z_t)] \leq \ln(C) + \sum_{i=0}^{k\ell-1} \left( \prod_{j=0}^{i} \pi_j \right) \ln\left( \frac{1}{\pi_i} \right) \tag{71}$$

Define $f_\pi(\sigma) = \pi\left( \ln\left( \frac{1}{\pi} \right) + \sigma \right)$. Then,

$$\sum_{i=0}^{k\ell-1} \left( \prod_{j=0}^{i} \pi_j \right) \ln\left( \frac{1}{\pi_i} \right) = \pi_0 \ln\left( \frac{1}{\pi_0} \right) + \pi_0 \pi_1 \ln\left( \frac{1}{\pi_1} \right) + \cdots + \pi_0 \pi_1 \ldots \pi_{k\ell-1} \ln\left( \frac{1}{\pi_{k\ell-1}} \right) \tag{72}$$

$$= f_{\pi_0}(f_{\pi_1}(\ldots f_{\pi_{k\ell-2}}(f_{\pi_{k\ell-1}}(0))\ldots)) \tag{73}$$

Note that if $\sigma \leq 1$, then $f_\pi(\sigma) \leq 1$ for any $0 \leq \pi \leq 1$. So, by induction,

$$\sum_{i=0}^{k\ell-1} \left( \prod_{j=0}^{i} \pi_j \right) \ln\left( \frac{1}{\pi_i} \right) \leq 1 \tag{74}$$

and we get the result. $\square$

Intuitively, if an algorithm can only increase $P(Z_t)$ (in expectation) by $1 + \ln(C)$ for every iteration and the algorithm isn't done until $P(Z_t) \geq k\ell \ln\left( \frac{C}{\lambda} \right)$, we have a lower bound.

**Lemma 9.** *For any algorithm that returns a subset of size at most $\lambda k$ at random iteration $\tau$,*

$$\Pr\left[ \tau \leq \left\lfloor \frac{k\ell \ln(C/\lambda)}{2(1 + \ln(C))} \right\rfloor \right] \leq \frac{1}{2} \tag{75}$$

*Proof.* Noting that $P(Z_1) = 0$, for any $T$,

$$\mathbb{E}[P(Z_{T+1})] = \sum_{t=1}^{T} \mathbb{E}[P(Z_{t+1} - P(Z_t)] \tag{76}$$

$$\leq T(1 + \ln(C)) \tag{77}$$

By Markov's inequality,

$$\Pr(\tau \leq T) \leq \Pr\left( P(Z_{T+1}) \geq k\ell \ln\left( \frac{C}{\lambda} \right) \right) \tag{78}$$

$$\leq \frac{\mathbb{E}[P(Z_{T+1})]}{k\ell \ln\left( \frac{C}{\lambda} \right)} \tag{79}$$

$$\leq \frac{T(1 + \ln(C))}{k\ell \ln\left( \frac{C}{\lambda} \right)} \tag{80}$$

If we set $T = \lfloor \frac{k\ell \ln(C/\lambda)}{2(1+\ln(C))} \rfloor$, then $\Pr(\tau \leq T) \leq \frac{1}{2}$. Or equivalently,

$$\Pr\left( \tau \geq \frac{k\ell \ln(C/\lambda)}{2(1 + \ln(C))} \right) \geq \frac{1}{2} \tag{81}$$

$\square$

Finally, we convert this bound into something more usable.

Note that $d^* = k$, $m = kC^\ell$ and $N = k\ell + 1$. Thus, $\ell = \frac{\ln(m/k)}{\ln(C)}$.

Set $C = \lceil e\lambda \rceil$.

Then, with probability at least $1/2$, $\tau$ is at least,

$$\frac{k\ell \ln(C/\lambda)}{2(1 + \ln(C))} \geq \frac{k\ln(m/k)\ln(e)}{2(1 + \ln(e\lambda + 1))(\ln(e\lambda + 1))} \tag{82}$$

$$\geq \frac{k\ln(m/k)}{2(2 + \ln(\lambda) + 1/e))(1 + \ln(\lambda) + 1/e)} \tag{83}$$

$$\geq \frac{k\ln(m/k)}{2(3 + \ln(\lambda))^2} \tag{84}$$

$$= \frac{d^*\ln(m/d^*)}{2(3 + \ln(\lambda))^2} \tag{85}$$

If $\lambda \leq \lambda' \log(m)\log(N) \leq \lambda' \log(m)\log(k\ln(m/k))$ (for constant $\lambda'$), then, the expected number of iterations is $\widetilde{\Omega}(d^* \log(m/d^*))$.

## F  ERROR SQUASHING

### F.1  BACKGROUND

In this subsection, we review related theoretical work in more detail as background to our framework and algorithms.

#### F.1.1  SETCOVER AND INTERACTIVE VARIANTS

Setcover is a classic computer science problem (Karp, 1972). While there are many similar and equivalent formulations, for this work, we formulate the problem in terms of $m$ binary decision variables and $N$ elements, where each element is represented by the decision variables that would "cover" the element: $Z_i \subset [m]$ for each $i \in [N]$. A solution is a subset $S \subset [m]$ such that $\forall i \in [N] : \exists s \in S : s \in Z_i$. In other words, we choose a subset $S \subset [m]$ of the decision variables so that all elements $Z_i$ intersect $S$. We wish to find a solution of smallest size $|S|$.

In the standard non-interactive version of setcover, all elements and sets are known, and there exist algorithms that return a solution of size $O(\log(N)C_{\text{OPT}})$ where $C_{\text{OPT}}$ is the size of the optimal solution set. In an interactive variant known as *online set cover* (Alon et al., 2009), the elements are not known at the beginning, only the number of decision variables $m$. The algorithm is initialized with $S_0 = \emptyset$. Then, for each iteration $t = 1, 2, \ldots$, an adversary chooses an element $i_t \in [N]$ that is not intersected by $S_t$ (i.e. $S_t \cap Z_{i_t} = \emptyset$) and reveals the decision variables that would cover $i_t$ (i.e. $Z_{i_t}$). The algorithm then chooses a set $s_t$ such that $s_t \in Z_{i_t}$ is permanently added to the solution: $S_t = S_{t-1} \cup \{s_t\}$. This process continues until all elements are covered and the adversary has no possible elements to choose. Note that the online set cover problem is harder than the standard set cover problem: if we were given all elements, we can simulate an adversary. Furthermore, the number of iterations $t$ is exactly the size of the final set. Alon et al. (2009) provides an algorithm that returns a solution of size $O(\log(N)\log(m)C_{\text{OPT}})$ where $C_{\text{OPT}}$ is the optimal (offline) solution.

In this work, we examine another interactive set cover variant that is harder than standard set cover but is a relaxed version of online set cover. We call this *exploratory set cover*. Like online set cover, the elements are not known at the beginning; the algorithm is only given the number of decision variables $m$. The algorithm has access to an adversarial oracle which takes a potential solution $S_t \subset [m]$ and reveals either that $S_t$ is a valid solution, or chooses an element $i_t$ such that $Z_{i_t}$ has an empty intersection with $S_t$ and reveals $Z_{i_t}$. However, unlike the online set cover version, the sets $S_t$ are arbitrary; for example, they need not be nested. At any iteration, the algorithm can return a final solution $\hat{S}$ that must intersect all $Z_i$. An algorithm is evaluated by two evaluation aspects, the number of exploratory iterations and the size of the final solution $|\hat{S}|$.

### F.1.2 SET COVER HARDNESS

Set cover is one of Karp's 21 NP-hard problems (Karp, 1972). Additionally, approximating set cover with an approximation factor of $o(\log n)$ is computationally hard. To be specific, unless NP problems can be solved in $n^{O(\log \log n)}$ time (a slightly weaker statement than P = NP), there is no algorithm for set cover that always returns a set cover of size $(1 - \epsilon) \ln(N) C_{\text{OPT}}$, where $C_{\text{OPT}}$ is the optimal set cover size (Feige, 1998). Alon et al. (2009) shows that approximating online set cover requires a larger approximation factor; any algorithm must return solutions of size at least $\widetilde{\Omega}(\ln(N) \ln(m) C_{\text{OPT}})$ where $C_{\text{OPT}}$ is the optimal (offline) solution (Alon et al., 2009). Note that the exploratory set cover problem is harder than standard set cover but easier than online set cover.

### F.1.3 MACHINE TEACHING

Machine teaching (Shinohara & Miyano, 1991; Goldman & Kearns, 1995) is a machine learning task where a teacher provides data points to the learner. In the most classic setting, the learner has a finite hypothesis class $\mathcal{H}$, where each hypothesis is a mapping from $\mathcal{X}$ to $\mathcal{Y}$, the teacher knows the learner's hypothesis class and which hypothesis is correct, and provides a teaching set $S \subset \mathcal{X}$ which uniquely determines the correct hypothesis. In particular, for a hypothesis class $\mathcal{H}$ and a correct hypothesis $h^*$, a set $S$ is a teaching set if

$$\forall h \in \mathcal{H} \setminus \{h^*\} : \exists x \in S : h(x) \neq h^*(x) \tag{86}$$

Intuitively, if the teacher provides $\{(x, h^*(x)) : x \in S\}$ to the learner, then the only remaining consistent hypothesis is $h^*$.

Machine teaching in this setting, where the teacher knows the entire hypothesis class of the learner, has a straightforward reduction to set cover. In particular, identify each hypothesis $h \in \mathcal{H} \setminus \{h^*\}$ as an element to be covered and

$$E_h = \{x \in \mathcal{X} : h(x) \neq h^*(x)\} \tag{87}$$

as the decision variables (datapoints in subset) that cover $h$.

An important quantity is the size of the smallest teaching set $d^*$, which is a function of both the hypothesis class $\mathcal{H}$ and the true hypothesis $h^* \in \mathcal{H}$. Note that the "teaching dimension" (Goldman & Kearns, 1995), is the maximum of $d^*$ over all $h^* \in \mathcal{H}$.

For the general non-realizable setting, Cicalese et al. (2020) defines a $k$-extended teaching set. For a hypothesis class $\mathcal{H}$ and true hypothesis $h^*$, if $h^*$ makes $k$ errors on a dataset, then $S$ is a $k$-extended teaching set if $|S \cap C_h| \geq k + 1$ for any hypothesis that makes at least $k + 1$ errors.

### F.2 INVARIANCE TO CONSISTENT ADDITIONS

Suppose we have a hypothesis class $\mathcal{H}$ and a learner $L$, where each $h \in \mathcal{H}$ is a function $h : \mathcal{X} \to \mathcal{Y}$ and the learner is a function:

$$L : \mathcal{P}(\mathcal{X} \times \mathcal{Y}) \to \mathcal{H} \tag{88}$$

Throughout this work, we make the following assumption about the learner.

**Definition 4** (Invariant to consistent additions). *We say a learner is invariant to consistent additions if $\forall D \subset \mathcal{P}(\mathcal{X} \times \mathcal{Y})$, $x \subset \mathcal{X}$, $y \subset \mathcal{Y}$:*

$$L(D)(x) = y \implies L(D) = L(D \cup \{(x, y)\}) \tag{89}$$

A simple example of a learner that meets this condition is a "ranked minimal error learner". To avoid handling infinite hypothesis classes, assume $\mathcal{X}$ is finite for this example. Fix a hypothesis class $\mathcal{H}$

(note $|\mathcal{H}| \leq |\mathcal{Y}|^{|\mathcal{X}|}$) and a bijective ranking $\sigma : \mathcal{H} \to [|\mathcal{H}|]$. A ranked minimal error learner returns the highest ranked hypothesis among the hypotheses in $\mathcal{H}$ achieving minimal error on the dataset:

$$L(D) = \underset{h:|\{(x,y) \in D: h(x) \neq y\}| = \min_{h' \in \mathcal{H}} |\{(x,y) \in D: h'(x) \neq y\}|}{\arg\min} \sigma(h) \tag{90}$$

**Proposition 1.** *A ranked minimal error learner is invariant to consistent additions.*

The proof of Proposition 1 can be found in Appendix F.6. The invariance to consistent additions is more general than ranked minimal error learners: see an example in Appendix F.7.1.

### F.3  INDUCED HYPOTHESIS CLASSES

Fix an size $m$ dataset $\{(x_i, y_i)\}_{i=1}^{m} \subset (\mathcal{X} \times \mathcal{Y})^m$. Note that the learner might have a very large hypothesis class, maybe even large enough to fit arbitrary labels for a $m$-sized dataset ($|\mathcal{H}| = |\mathcal{Y}|^m$). However, we can define the *induced hypothesis class* as

$$\overline{\mathcal{H}} = \{L(\{(x_i, y_i) : i \in S\}) : S \subset [m]\} \tag{91}$$

which could be significantly smaller, depending on the structure of the model and data. Then, we can define the *error sets* as

$$\mathcal{E} = \{\{i : h(x_i) \neq y_i\} : h \in \overline{\mathcal{H}}\} - \{\emptyset\} \tag{92}$$
$$\tag{93}$$

where $E_h \in \mathcal{E}$ is the set of datapoints that eliminate a hypothesis $h$.

For notational convenience, define $L^{(\mathrm{err})} : \mathcal{P}([m]) \to \mathcal{P}([m])$ as the set of errors when trained on a subset $S$:

$$L^{(\mathrm{err})}(S) = \{i : L(\{(x_{i'}, y_{i'}) : i' \in S\})(x_i) \neq y_i\} \tag{94}$$

then, $\mathcal{E}$ is simply,

$$\mathcal{E} = \{L^{\mathrm{err}}(S) : S \subset [m]\} - \{\emptyset\} \tag{95}$$

### F.4  ERROR INCLUSIVITY

We wish to find a set $S$ such that makes as few errors as if we trained on the entire dataset: $|L^{(\mathrm{err})}(S)| = |L^{(\mathrm{err})}([m])|$. However, this condition is a bit general as we might happen to have $|L^{(\mathrm{err})}(\emptyset)| = |L^{(\mathrm{err})}([m])|$, in which case $S = \emptyset$ is perhaps unlikely to generalize to test points. Instead, we might wish to find a set $S$ that, when trained on, yields no errors outside of $S$.

**Definition 5.** *For a fixed dataset, we say a set is "error inclusive" if the following holds:*

$$L^{(err)}(S) \subset S \tag{96}$$

Note that $[m]$ is always error inclusive. Furthermore, if a learner is invariant to consistent additions, $L^{(\mathrm{err})}(S) = L^{(\mathrm{err})}([m])$.

**Proposition 5.** *Suppose a learner $L$ is invariant to consistent additions. If a set $S$ is error inclusive,*

$$L^{(err)}(S) = L^{(err)}([m]) \tag{97}$$

The proof of Proposition 5 is in Appendix F.6. Interpreted further, $|L^{(\mathrm{err})}(S)| = |L^{(\mathrm{err})}([m])|$ and $L^{(\mathrm{err})}([m]) \subset S$, meaning that $S$ contains the errors of training on the full dataset.

### F.4.1 INTERPOLATING LEARNERS

We say a learner is "interpolating" if it makes no training errors: $L^{(\mathrm{err})}(S) \cap S = \emptyset$. In this case, we can achieve 0 errors on the entire dataset. Furthermore, note that for interpolating learners, error inclusivity of a set $S$ implies that training on $S$ results in zero error on the entire dataset.

### F.4.2 RESULTS

Define the *extended error sets* as

$$\mathcal{E}^+ = \{L^{(\mathrm{err})}(S) \setminus S : S \subset [m]\} - \{\emptyset\} \tag{98}$$

**Proposition 3.** *Fix a dataset. If a learner is invariant to consistent additions,*

$$S \text{ fully intersects } \mathcal{E}^+ \iff S \text{ is error inclusive} \tag{99}$$

The proof can be found in Appendix F.6.

Define

$$d^* = \min_{S \text{ fully intersects } \mathcal{E}^+} |S| \tag{100}$$

Note that in the case of interpolating learners, $\mathcal{E}^+ = \mathcal{E}$, and $d^*$ is the size of the optimal teaching set.

Thus, as in the machine teaching case, we have reduced the problem of finding a good subset to finding a minimal setcover, though of an expanded set.

Note that we can convert a data subset selection algorithm with interpolating learners (online or exploratory set cover algorithm) into a subset selection algorithm without assuming an interpolating learner, simply by acting as if training errors don't exist. In particular, if we train on a subset $S_t$ and receive a hypothesis $h_t$, rather than intersecting $\{i : h_t(x_i) \neq y_i\}$, we can intersect $\{i : h_t(x_i) \neq y_i \wedge i \notin S_t\}$. Put another way, when using a realizable algorithm on a non-interpolating learner, we can "squash" the training errors and pretend they don't exist. In fact, this technique is used as a heuristic in Cicalese et al. (2020) to make the algorithm work in practice. Thus, we provide a framework justifying the use of this technique previously used as a heuristic.

### F.5 COMPARISON TO K-EXTENDED TEACHING DIMENSION

Cicalese et al. (2020) introduces the concept of $k$-extended teaching set for a hypothesis class $\mathcal{H}$ and a dataset $\{(x_i, y_i)\}_{i=1}^m \subset (\mathcal{X} \times \mathcal{Y})^m$. In particular, let $k = \min_{h \in \mathcal{H}} |\{i \in [m] : h(x_i) \neq y_i\}|$ be the minimal number of errors of a hypothesis in $\mathcal{H}$. A set $S$ is a $k$-extended teaching set, if for any $h \in \mathcal{H}$ such that $|\{i \in [m] : h(x_i) \neq y_i\}| \geq k+1, |\{i \in S : h(x_i) \neq y_i\}| \geq k+1$. In other words, any hypothesis that makes $k+1$ errors (more than the optimal number) on the entire dataset, makes $k+1$ errors on the selected subset $S$.

Here, for ranked minimal error learners, we compare $k$-extended teaching sets to error inclusive sets (and thus sets that fully intersect $\mathcal{E}^+$). Fix a dataset, hypotheses $\mathcal{H}$, and ranking $\sigma$. Let $k$ be the minimal number of errors for a hypothesis $h \in \mathcal{H}$ for $\vec{D}$. For a ranked minimal error learner, $|L^{(\mathrm{err})}([m])| = k$, so an error inclusive set must make $k$ errors. Therefore any hypothesis $h$ where $\sigma(h) < \sigma(L(\{(x_i, y_i) : i \in [m]\})) = \sigma^*$ must incur at least $k+1$ errors on $S$, and any hypothesis $h$ where $\sigma(h) > \sigma(L(\{(x_i, y_i) : i \in [m]\})) = \sigma^*$ must incur at least $k$ errors on $S$. A comparison of the two concepts is shown in Table 5.

Note that error inclusivity requires one less error on Type II hypotheses and $k$ more errors on Type III (error optimal) hypotheses. We can construct cases where the smallest error inclusive subset is of size $m$ but the smallest $k$-extended teaching set is of size 0 (see Appendix F.7.3). In the other direction, there are cases where the smallest error inclusive subset is of size 2 while the smallest $k$-extended teaching set is of size $m-1$ (see Appendix F.7.4). Thus, the two notions of optimality are similar but neither is a stronger condition than the other.

Table 5: A comparison between $k$-extended teaching sets and error inclusive sets. The three columns correspond to three types of hypotheses separated by two attributes: whether a hypothesis is ranked higher or lower than $L^{(\text{err})}([m])$, and whether the total number of errors for a hypothesis is optimal, $k$, or sub-optimal, $\geq k+1$. Note that $L^{(\text{err})}([m])$ is the highest ranked hypothesis with optimal errors $k$, so there are no hypotheses with $k$ errors and rank lower than $\sigma^*$.

| Hypothesis type | Type I | Type II | Type III |
|---|---|---|---|
| $\sigma(h)$ | $< \sigma^*$ | $> \sigma^*$ | $> \sigma^*$ |
| Total errors | $\geq k+1$ | $\geq k+1$ | $= k$ |
| Errors for $k$-extended | $\geq k+1$ | $\geq k+1$ | $\geq 0$ |
| Errors for error inclusive | $\geq k+1$ | $\geq k$ | $\geq k$ |

## F.6 FRAMEWORK PROOFS

**Proposition 1.** *A ranked minimal error learner is invariant to consistent additions.*

*Proof.* Suppose $L$ is a ranked minimal error learner with hypothesis class $\mathcal{H}$ and ranking $\sigma$. Fix a dataset $D \subset \mathcal{P}(\mathcal{X} \times \mathcal{Y})$, an input $x \in \mathcal{X}$, and an output $y \in \mathcal{Y}$. It suffices to show that $L(D)(x) = y$ implies $L(D) = L(D \cup (x, y))$.

Define $D^+ = D \cup \{(x, y)\}$. Define $E : \mathcal{P}(\mathcal{X} \times \mathcal{Y}) \times \mathcal{H} \to \mathbb{Z}_+$ as $E(D, h) = |\{(x, y) \in D : h(x) \neq y\}|$.

$$\underline{E} = \min_{h \in \mathcal{H}} E(D, h) \tag{101}$$

$$\underline{E}^+ = \min_{h \in \mathcal{H}} E(D^+, h) \tag{102}$$

$$M = \{h : E(D, h) = \underline{E}\} \tag{103}$$

$$M^+ = \{h : E(D^+, h) = \underline{E}^+\} \tag{104}$$

note that

$$L(D) = \arg \min_{h \in M} \sigma(h) \tag{105}$$

$$L(D^+) = \arg \min_{h \in M^+} \sigma(h) \tag{106}$$

Define

$$\underline{\sigma} = \min_{h \in M} \sigma(h) \tag{107}$$

$$\underline{\sigma}^+ = \min_{h \in M} \sigma(h) \tag{108}$$

Because $L(D) \in M$, $E(D, L(D)) = \underline{E}$.

Because $L(D)(x) = y$ and $E(D, L(D)) = \underline{E}$, $E(D^+, L(D)) = \underline{E}$, so $\underline{E}^+ \leq \underline{E}$.

Because $D \subset D^+$, so $\underline{E}^+ \geq \underline{E}$.

Therefore, $\underline{E}^+ = \underline{E}$.

Because $E(D, h) \leq E(D^+, h)$ and $\underline{E}^+ = \underline{E}$, $M^+ \subset M$. Furthermore, $E(D^+, L(D)) = \underline{E} = \underline{E}^+$, so $L(D) \in M^+$.

$$\sigma(L(D)) = \min_{h \in M} \sigma(h) \leq \min_{h \in M^+} \sigma(h) \leq \sigma(L(D)) \tag{109}$$

Because $\sigma$ is bijective, it has a unique argmax.

so $L(D^+) = \arg\min_{h \in M^+} \sigma(h) = L(D)$. □

**Lemma 10.** *Suppose a learner $L$ is invariant to consistent additions. Then, for a fixed dataset $\{(x_i, y_i) : i \in [m]\}$, and for any subsets $S, S' \subset [m]$,*

$$L^{(err)}(S) \cap S' = \emptyset \implies L^{(err)}(S \cup S') = L^{(err)}(S) \tag{110}$$

*Proof.* We prove by induction on the size of $S'$.

As the base case, if $|S'| = 0$, the statement trivially holds.

Suppose the lemma holds for any $|S'| = k$. Now, we must show $L^{(err)}(S) \cap S' = \emptyset \implies L^{(err)}(S \cup S') = L^{(err)}(S)$ for $|S'| = k + 1$.

Choose $s$ and $S''$ with $|S''| = k$ such that $S' = S'' \cup \{s\}$. Then, $L^{(err)}(S) \cap S' = \emptyset$ implies $L^{(err)}(S) \cap S'' = \emptyset$ and thus $L^{(err)}(S \cup S'') = L^{(err)}(S)$.

Furthermore, $L^{(err)}(S) \cap S' = \emptyset$ implies $s \notin L^{(err)}(S) = L^{(err)}(S \cup S'')$. Therefore,

$$L(\{(x_i, y_i) : i \in S \cup S''\})(x_s) = y_s \tag{111}$$

since the learner is invariant to consistent additions,

$$L(\{(x_i, y_i) : i \in S \cup S''\}) = L(\{(x_i, y_i) : i \in S \cup S''\} \cup \{(x_s, y_s)\}) = L((x_i, y_i) : i \in S \cup S'' \cup \{s\}\}) \tag{112}$$

$$L^{(err)}(S) = L^{(err)}(S \cup S'') = L^{(err)}(S \cup S'' \cup \{s\}) = L^{(err)}(S \cup S') \tag{113}$$

□

**Proposition 5.** *Suppose a learner $L$ is invariant to consistent additions. If a set $S$ is error inclusive,*

$$L^{(err)}(S) = L^{(err)}([m]) \tag{114}$$

*Proof.* Note that since $L^{(err)}(S) \subset S$, $L^{(err)}(S) \cap ([m] \setminus S) = \emptyset$.

By Lemma 10, this implies $L(S) = L(S \cup ([m] \setminus S)) = L([m])$. Then, $L^{(err)}(S) = L^{(err)}([m])$. □

**Lemma 11.** *Suppose a learner $L$ is invariant to consistent additions. For any sets $S^*, S \subset [m]$, if $S^*$ is error inclusive, then either*

$$|(L^{(err)}(S) \setminus S) \cap S^*| \geq 1 \tag{115}$$

*or*

$$L^{(err)}(S^*) = L^{(err)}(S) \tag{116}$$

*Proof.* By the assumption of error inclusivity, $L^{(err)}(S^*) \setminus S^* = \emptyset$.

Thus $L^{(err)}(S^*) \cap \neg S^* \cap S = \emptyset$ and $L^{(err)}(S^*) \cap (S \setminus S^*) = \emptyset$. Since $L$ is invariant to consistent additions, $L^{(err)}(S^*) = L^{(err)}(S^* \cup (S \setminus S^*)) = L^{(err)}(S \cup S^*)$.

Suppose the first conclusion is not satisfied: $(L^{(err)}(S) \setminus S) \cap S^* = \emptyset$, then $L^{(err)}(S) \cap (S^* \setminus S) = \emptyset$. Since the learner is invariant to consistent additions, $L^{(err)}(S) = L^{(err)}(S \cup (S^* \setminus S)) = L^{(err)}(S \cup S^*)$.

Thus, $L^{(\mathrm{err})}(S) = L^{(\mathrm{err})}(S \cup S^*) = L^{(\mathrm{err})}(S^*)$.

$\square$

**Proposition 3.** *If a learner $L$ is invariant to consistent additions, then,*

$$S \text{ fully intersects } \mathcal{E}^+ \iff S \text{ is error inclusive} \tag{117}$$

*Proof.* $\rightarrow$: Suppose $S$ fully intersects $\mathcal{E}^+$. Then, for any set $S'$, either $L^{(\mathrm{err})}(S') \setminus S' = \emptyset$ or $|(L^{(\mathrm{err})}(S') \setminus S') \cap S| \geq 1$.

Set $S' = S$. Then, because $(L^{(\mathrm{err})}(S) \setminus S) \cap S = \emptyset$, it must be the case that $L^{(\mathrm{err})}(S) \setminus S = \emptyset$ and thus $S$ is error inclusive.

$\leftarrow$: Suppose a set $S$ is error inclusive.

Pick any set $S' \subset [m]$. It suffices to show that either $L^{(\mathrm{err})}(S')/S' = \emptyset$ or $|(L^{(\mathrm{err})}(S')/S') \cap S| \geq 1$.

By Lemma 11, there are two cases:

**Case 1:** $|(L^{(\mathrm{err})}(S') \setminus S') \cap S| \geq 1$.

We are done.

**Case 2:** $L^{(\mathrm{err})}(S) = L^{(\mathrm{err})}(S')$ and $(L^{(\mathrm{err})}(S') \setminus S') \cap S = \emptyset$.

$$(L^{(\mathrm{err})}(S') \setminus S') \cap S = \emptyset \tag{118}$$
$$(L^{(\mathrm{err})}(S') \cap S) \cap \neg S' = \emptyset \tag{119}$$
$$(L^{(\mathrm{err})}(S) \cap S) \cap \neg S' = \emptyset \tag{120}$$
$$L^{(\mathrm{err})}(S) \cap \neg S' = \emptyset \tag{121}$$
$$L^{(\mathrm{err})}(S') \cap \neg S' = \emptyset \tag{122}$$
$$L^{(\mathrm{err})}(S') \setminus S' = \emptyset \tag{123}$$
$$\tag{124}$$

where the third to last line follows from the error inclusivity of $S$.

$\square$

## F.7 EXAMPLES

### F.7.1 EXAMPLE OF LEARNER THAT IS NOT A MINIMAL ERROR LEARNER, BUT IS INVARIANT TO CONSISTENT ADDITIONS

We examine a situation with three datapoints and binary labels. The dataset is $D = \{(x_1, 0), (x_2, 0), (x_3, 0)\}$ and the learner $L$ has two hypotheses $h_1$ and $h_2$ where

$$h_1(x_1) = 1 \tag{125}$$
$$h_1(x_2) = 0 \tag{126}$$
$$h_1(x_3) = 0 \tag{127}$$
$$h_2(x_1) = 0 \tag{128}$$
$$h_2(x_2) = 1 \tag{129}$$
$$h_2(x_3) = 1 \tag{130}$$

the learner has the following mapping:

$$L(\emptyset) = h_2 \tag{131}$$
$$L(\{(x_1, 0)\}) = h_2 \tag{132}$$
$$L(\{(x_2, 0)\}) = h_1 \tag{133}$$
$$L(\{(x_3, 0)\}) = h_1 \tag{134}$$
$$L(\{(x_1, 0), (x_2, 0)\}) = h_2 \tag{135}$$
$$L(\{(x_1, 0), (x_3, 0)\}) = h_2 \tag{136}$$
$$L(\{(x_2, 0), (x_3, 0)\}) = h_1 \tag{137}$$
$$L(\{(x_1, 0), (x_2, 0), (x_3, 0)\}) = h_2 \tag{138}$$

Note that $h_2$ has two errors on $D$, while $h_1$ only has one error. However, $L(D) = h_2$. Thus, the learner does not return the hypothesis with minimal error.

However, a simple examination yields invariance to consistent additions. For example, $L(\{(x_2, 0)\})(x_3) = y_3$ and $L(\{(x_2, 0)\}) = L(\{(x_2, 0), (x_3, 0)\})$.

### F.7.2 COMPARISON BETWEEN INDUCED HYPOTHESIS CLASS AND EXTENDED ERROR SETS

Here we show that $\mathcal{E}^+$ can be larger than $\mathcal{E}$.

We examine a situation with three datapoints and binary labels. The dataset is $D = \{(x_1, 0), (x_2, 0), (x_3, 0)\}$ and the learner $L$ has two hypotheses $h_1$ and $h_2$ where

$$h_1(x_1) = 1 \tag{139}$$
$$h_1(x_2) = 0 \tag{140}$$
$$h_1(x_3) = 0 \tag{141}$$
$$h_2(x_1) = 0 \tag{142}$$
$$h_2(x_2) = 1 \tag{143}$$
$$h_2(x_3) = 1 \tag{144}$$

$$L(\emptyset) = h_2 \tag{145}$$
$$L(\{(x_1, 0)\}) = h_2 \tag{146}$$
$$L(\{(x_2, 0)\}) = h_1 \tag{147}$$
$$L(\{(x_3, 0)\}) = h_1 \tag{148}$$
$$L(\{(x_1, 0), (x_2, 0)\}) = h_2 \tag{149}$$
$$L(\{(x_1, 0), (x_3, 0)\}) = h_2 \tag{150}$$
$$L(\{(x_2, 0), (x_3, 0)\}) = h_1 \tag{151}$$
$$L(\{(x_1, 0), (x_2, 0), (x_3, 0)\}) = h_1 \tag{152}$$

Then, $\mathcal{E}^+ = \{\{1\}, \{2, 3\}, \{2\}, \{3\}\}$ while $\mathcal{E} = \{\{1\}, \{2, 3\}\}$.

### F.7.3 K-EXTENDED TEACHING SET SUPERIORITY

In the following example, $\mathcal{H} = \{h_i\}_i$ and $\sigma(h_i) = i$.

$$y_. = (0, 0, 0, \ldots, 0, 0) \tag{153}$$
$$h_1(x_.) = (1, 0, 0, 0, 0, \ldots, 0, 0) \tag{154}$$
$$h_2(x_.) = (0, 1, 0, 0, 0, \ldots, 0, 0) \tag{155}$$
$$h_3(x_.) = (0, 0, 1, 0, 0, \ldots, 0, 0) \tag{156}$$
$$h_4(x_.) = (0, 0, 0, 1, 0, \ldots, 0, 0) \tag{157}$$
$$\ldots \tag{158}$$
$$h_m(x_.) = (0, 0, 0, 0, 0, \ldots, 0, 1) \tag{159}$$

In this case, the smallest 1-extended teaching set is $\emptyset$ while the smallest error inclusive set is $\{1, 2, \ldots, m\}$.

### F.7.4 ERROR INCLUSIVITY SUPERIORITY

In the following example, $\mathcal{H} = \{h_i\}_i$ and $\sigma(h_i) = i$.

$$y_. = (0, 0, 0, 0, 0, \ldots, 0, 0, 0) \tag{160}$$
$$h_1(x_.) = (1, 0, 0, 0, 0, \ldots, 0, 0, 0) \tag{161}$$
$$h_2(x_.) = (0, 1, 0, 0, 0, \ldots, 0, 0, 1) \tag{162}$$
$$h_3(x_.) = (0, 0, 1, 0, 0, \ldots, 0, 0, 1) \tag{163}$$
$$h_4(x_.) = (0, 0, 0, 1, 0, \ldots, 0, 0, 1) \tag{164}$$
$$\ldots \tag{165}$$
$$h_{m-1}(x_.) = (0, 0, 0, 0, 0, \ldots, 0, 1, 1) \tag{166}$$

In this case, the smallest 1-extended teaching set is $\{2, 3, \ldots, m\}$ while the smallest error inclusive set is $\{1, m\}$.

