# OpenReview forum: "Data Subset Selection via Machine Teaching"
_ICLR.cc/2023/Conference — Submitted to ICLR 2023_

### Official Review · Reviewer_3ZRi · 2022-10-23

**Confidence:** 3
**Clarity, Quality, Novelty And Reproducibility:** 1. The main algorithm proposed employ…
**Correctness:** 3
**Technical Novelty And Significance:** 3
**Empirical Novelty And Significance:** 2
**Recommendation:** 5

**Strength And Weaknesses:**

Strengths:
1. A solid theoretical analysis of the algorithm is given, alongside a detailed theoretical section for justifying the use of error squashing.
2. Theorems and propositions are explained well with interpretations.
3. Rather extensive experiments are conducted across multiple baseline methods, datasets and model architectures.

Weaknesses:
1. The novelty of the paper needs further justification.

**Summary Of The Paper:**

This paper identifies theoretical gaps in the existing literature on data subset selection in machine teaching, and analysis the justification of the error-squashing heuristics adopted in the previous works. With that, the authors propose a data subset selection algorithm with near-optical guarantees on the query complexity and the size of the returned subset. The paper also shows empirical experiments on 6 datasets, 6 baseline methods and 3 model architectures to demonstrate the effectiveness of the proposed algorithm.

**Summary Of The Review:**

I am majorly concerned with the novelty of the paper. As discussed above, the algorithm mainly adopts existing techniques with an addition of a probability halving phase that allows drops in selection probability. The improvement in the theoretical guarantee of the algorithm is also likely due to this trick which essentially relaxes the problem setting. The second part on the theoretical understanding of error squashing is nice to have, but I wonder how much theoretical and empirical insights it could have to impact the community to move forward. The practicability of the method is also largely limited by its time and computational overhead. The authors could probably explain more about the significance of the theoretical understandings presented.

---

> ### Author Response · Authors · 2022-11-09
> **Author response**
>
> We would like to warmly thank you for your thorough review and constructive comments. We hope that we will succeed in convincing you of the usefulness of our work. Please also see our comment to all reviewers.
>
> 1. We agree that our work builds on the existing algorithms in Dasgupta et al. and Cicalese et al. and uses similar algorithmic pieces. Our theoretical contributions are not in the form of the algorithm we present (except the probability reduction, see 2.), but rather the analysis, showing near-optimal performance (via upper and lower bounds) and justifying error-squashing.
> 2. Removing points from the subsets is critical for our improved upper bound. See Section 4 of Alon et al. (2009) for a lower bound for the online set cover problem, which is equivalent to black-box machine teaching. Thus, no algorithm in the black-box machine teaching setting can achieve our near-optimal upper bound. This fact motivates the introduction of our setting (a relaxation of black-box machine teaching) where points can be removed from the queried subsets.
> 3. In our paper, we provide ranked minimal error learners as an example of learners that are invariant to consistent additions. This class of learners encompasses the classic consistent learners with a finite hypothesis class used in the machine teaching literature. We do not think the sentence with “does not change the learner’s prediction” is misleading but is rather an accurate informal description of the condition. Note that if two hypotheses make the same predictions everywhere, they are the same hypothesis (we define hypotheses as functions). See our response to question 8 for further discussion.
> 4. Thank you for pointing out the use of “for any” instead of “for all”. We have changed it to “for all” to reduce ambiguity.
> 5. Thank you for catching this typo. We have corrected it.
> 6. The phenomenon of random sampling performing well for small subset sizes is a well-documented phenomenon. For example, Guo et al. (2022) and Sorscher et al. (2022) show a similar result for data selection. From the related active learning literature, it is not uncommon for methods to be outperformed by random sampling. An explanation for this phenomenon is beyond the scope of this work.
> 7. The hyperparameters for the machine teaching techniques can be found in Table 2 of the appendix. From our hyperparameter tuning experiments, decreasing the hyperparameters causes bad performance (worse than random sampling) and increasing the hyperparameters creates larger subset sizes but does not decrease the pool error. Perhaps a good way to produce smaller subsets is to terminate early, when the pool error is below some threshold. Producing larger subsets would seem to require more knowledge than just the pool error.
> 8. We believe that “achieving near-zero pool error is insufficient to achieve minimal test error” for SVHN and FMNIST because the learners for these two datasets are not invariant to consistent additions. Note that if the learners for these two datasets were invariant to consistent additions, subsets that achieve zero pool error (or error inclusive sets) would yield the same hypotheses (and thus the same test performance) as training on the entire pool, similar to the results on the other four datasets.
> 9. We agree that the machine teaching methods as presented have a large overhead. However, there exist techniques, mostly variations of the “Select via Proxy” idea (Coleman et al., 2020), which can significantly decrease the computational cost of data selection. Some examples of proxies are training smaller models or training for fewer epochs.
>
> We look forward to your response and are available to answer any other questions.
>
>
>
> ## Additional references:
>
> Ben Sorscher, Robert Geirhos, Shashank Shekhar, Surya Ganguli, Ari S. Morcos. Beyond neural scaling laws: beating power law scaling via data pruning. NeurIPS 2022.

---

> > ### Comment · Reviewer_3ZRi · 2022-11-18
> > **Post rebuttal**
> >
> > I would like to thank the authors for their response and clarification. I will keep my score.

---

### Official Review · Reviewer_zPDn · 2022-10-25

**Confidence:** 4
**Correctness:** 3
**Technical Novelty And Significance:** 3
**Empirical Novelty And Significance:** 3
**Recommendation:** 6

**Clarity, Quality, Novelty And Reproducibility:**

I think the topic in this paper is interesting and this paper is generally well-written.

I have the following comments/questions. I look forward to the response/clarification from the author(s). Thanks.


Clarity:

1. At several places in this paper, it mentioned "zero error", I am not sure that it is reasonable. Generally speaking, in machine learning, "zero training error" may possibly cause "overfitting". Why use such a condition? Could you please explain that?

2. Some experimental details are not clearly given; for example, it mentioned that "the standard train/test splits" (On Page 7, Section 5, Line 2 in the 2nd Paragraph), could you please detail it?

3. For the results in Figures 1 and 2 (also, including the Supplementary results), from the description, I understood that error bars were from training with 10 replications on a subset. I checked them, especially the results from CFLM, DHPZ, and the proposed algorithm, and I found almost no error fluctuations (except a little fluctuation on FMNIST), I wonder if this indicated that the learning results of these three algorithms are very stable? And I'm curious about the design of 10 replications, the 10 replications is a 10 random train/test splits, right? Also, for these three algorithms might consider the performance with the changing of subset sizes?

4. For the results in Table 1 (also in Supplementary Tables 3 and 4), could you please explain them further?

5. If the input data matrix is transposed, does the method in this paper, i.e., data subset selection, become feature selection?

In addition, some other tiny issues/typos

(1) The format of the references is quite inconsistent (such as, sometimes the author's name is abbreviated, sometimes it is not; sometimes the first letter of every word of journal/conference names is capitalized, sometimes it is not; and so onThere are times when the first letter of each word is capitalized, sometimes not, and so on). Please check carefully and correct it.

(2) In Table 2, there is no punctuation at the end of the caption.

(3) It may not be necessary to number each mathematical formula in the text. In general, we only need to number the mathematical formula that will be used/cited later.

---



**Strength And Weaknesses:**

### Strength:

1. Selecting an informative subset from a dataset is significant in practice. An obvious advantage is that the training/computation time of algorithms based on such data can be largely reduced.

2. Theoretical analysis and empirical experiments are both performed for validating the proposed algorithm.

---

### Weaknesses:

1. Maybe some technique assumptions in this paper are not so reasonable.

2. Some descriptions (including experimental details) in this paper are not clear.

For more details, please see the section of "Clarity, Quality, Novelty And Reproducibility".


---



**Summary Of The Paper:**

In this paper, the author(s) considered the problem of data subset selection. The author(s) proposed a machine teaching subset selection algorithm with theoretical guarantees. Through empirical experiments, the author(s) showed the advantage of the proposed algorithm.

---

**Summary Of The Review:**

The work in this paper is interesting; However, there are some unclear aspects in the description (including experiments) of this paper. Maybe it needs the author(s) to clarify them.

In addition, in this paper, the author(s) performed empirical experiments, but I am not sure the empirical description/experiments would be enough to reproduce because no code of this paper seems to be provided. Thanks.

---

---

> ### Author Response · Authors · 2022-11-09
> **Author response**
>
> We are very grateful for the time you spent reading our paper and detailing your concerns. We hope that we will answer all your questions in a satisfactory manner.
>
> 1. Thank you for your question. We don’t actually require zero training error because of the “error-squashing” technique (see Sections 3.3 and 4.4 in our paper). Instead, for simplicity, we begin the exposition with the zero training error assumption and then extend it to the more general setting in Section 3.3. We would furthermore like to point out that generalization and overfitting for neural networks are not well-understood. For example, often the best performing neural networks achieve significantly lower training error than test error, similar to interpolating models. Furthermore,previous work has shown that common neural networks can fit any training labels, including random labels, and thus the usual story of overfitting and generalization is inadequate for neural networks.
> 2. Thank you for this clarification question. We have changed “standard” to “predefined”, meaning that all datasets we used have predefined train/test splits in the dataset download. Please see the documentation for “torchvision.datasets” for more information.
> 3. Your understanding of the replications is correct. To address training randomness for neural networks (batch order and random initialization), for each subset, we train the network ten independent times.
> 4. To describe Table 1 better, we have replaced “iteration” with “queries to the learner”. For example, the DHPZ method requires 147 queries to the learner for CIFAR10.
> 5. Transposing the data matrix is a very interesting idea to perform feature selection. While similar techniques could be used for feature selection, we don’t think transposing the data matrix alone would accomplish this since a hypothesis has a class prediction for each datapoint, but not for each feature.
>
> (1) Thank you for pointing out the inconsistencies in the references. We have, as much as possible, removed inconsistencies.
>
> (2) Thank you for pointing out this typo. We have corrected it.
>
> (3) Thank you for the comment on including equation numbers. We find that equation numbers are helping for referencing equations (e.g., in reading groups) even if they are not referenced by the paper itself. We are happy to make this change if you feel very strongly about this and the other reviewers agree.
>
> As is typical for papers with empirical contributions, we will release code upon de-anonymization of the paper. Currently, Github links would identify the authors.
>
> We eagerly await your response and are available to answer any other questions.

---

### Official Review · Reviewer_YuUz · 2022-10-25

**Confidence:** 4
**Correctness:** 3
**Technical Novelty And Significance:** 2
**Empirical Novelty And Significance:** 2
**Recommendation:** 5

**Clarity, Quality, Novelty And Reproducibility:**

Somewhat clear, primarily builds on Cicalese with some novel contributions and should be reproducible.

I should note the primary contribution is the theoretical justification for error squashing. I'm not sure about the other contribution stated in the introduction ("introduce algorithm").

**Strength And Weaknesses:**

Strengths
- Proves upper and lower bounds for performance of subset selection algorithm
- Good evaluation with 6 baselines tested alongside the introduced algorithm, tested on 6 datasets and ablated on 3 NN architectures.
- Provides theoretical analysis of error squashing which was missing from Cicalese et al 2020, when they introduced error squashing.

Weaknesses
- Seems to be a minor improvement on Cicalese et al. (2020); and the experimental results suggest the same-- with roughly the same performance (sometimes slightly worse and sometimes slightly better) as Cicalese's and Dasgupta's work.

Minor comments & questions
- Notation:
-- In section 2/3: use of '[m]' describe set of all pool points is non-standard and would be nice to keep to the nomenclature in the subfield (e.g. notation used in Cicalese 2020, SVP by Coleman 2020 or Kilmasetty 2020).
-- Switching of notation of error on hypothesis classes from err[h] to E interchangebly.
-- Is E_{t} just shorthand for E_{h_t}
- More clarity on improvements as compared to Cicalese
- Add standard deviation bounds to experimental results based on multiple runs.


**Summary Of The Paper:**

Paper presents theoretical and experimental analysis of a machine teaching algorithm for data subset selection.

**Summary Of The Review:**

Somewhat minor improvement, but some good theoretical and empirical analysis. Improvements to exposition would assist. I think this is below the threshold for acceptance due to the lack of significant novelty here.

---

> ### Author Response · Authors · 2022-11-09
> **Author response**
>
> Thank you for your review and expressing your concerns.
>
> If you can point out the “confusing and inconsistent” notation in sections 3 and 4, we would be very happy to address them.
>
> For our contributions and the improvement on Cicalese et al., please see our comment above to all reviewers. We would like to highlight that our empirical claim is that machine teaching algorithms perform well on standard data selection tasks, rather than that our algorithm is empirically superior to the other two machine teaching approaches.
>
> Could you clarify what you mean by “add bounds to plots”? As is typical in theoretical works and since we did not optimize constants, our results are certainly loose in terms of constants and we are not sure how meaningful adding them to the plots would be. Instead, the dependence on problem-specific quantities is typically the goal of theoretical analyses.
>
> Please let us know if you have additional concerns or questions as we stand ready to answer.

---

> > ### Comment · Reviewer_YuUz · 2022-12-07
> > **Response to rebuttal**
> >
> > Dear Authors,
> >
> > My apologies for the late response and lack of engagement. I've noted some of the confusing notation in my updated original review. I also note that I've moved that concern to the minor comments section.
> >
> > Regarding the contributions:
> > - I think the contribution of proposing a machine teaching subset selection algorithm is overstated. Some additional work on the theoretical bound was added as compared to Cicalese.
> > - The analysis of error squashing is welcomed
> > - The application to image datasets is again welcome but I don't think it's a particularly big contribution.
> >
> > If I had to frame the paper it has two parts: (1) analysis of error squashing in the context of machine teaching subset selection algorithms, with a novel lower bound of performance; (2) empirical results of machine teaching subset selection on image datasets with NNs.
> >
> > I think the disconnect between the reviewers and the authors is a result of the framing this paper as introducing a novel machine subset selection algorithm. I think had the contributions been stated more conservatively and the paper framed as bringing new understand to error squashing it would have been more warmly received.
> >
> > I'll add that it feels like there's a missing experiment set and then exposition: showing that DHPZ (Dasgupta) fails without error-squashing and what it would look like with/out error-squashing both in their experimental set-ups and this paper's experimental set-up.
> >
> > On the "bounds." Typo on my part--my apologies again. I simply meant standard deviation bounds. I thought they weren't present in many of the plots, but in fact it just seems like the standard deviation was just very small? Although both figure 1 and 2 look very suspicious as the bounds are far tighter than I've seen with my own subset selection experiments. Nonetheless, I take the authors word for it but would have appreciated the code being provided via an anonymous repo or zip file.
> >
> > At this stage, give the framing of paper and the contributions therein I'm inclined to reject the paper and retain my score of 5 for now.

---

### Author Response · Authors · 2022-11-09
**Author response to all reviewers**

We would like to thank all of the reviewers for their time reading our paper and writing thoughtful reviews. It seems that all reviewers are in agreement about the **quality of the theoretical and empirical work**. Reviewers YuUz and zPDn had **concerns about the clarity of exposition**. We have updated the paper and posted individual responses to the reviewers to address these concerns. Reviewers YuUz and 3ZRi questioned **the significance of this paper**. In addition to individual responses, we would like to try to convince the reviewers of the importance of our work via a summary of our contribution below.

Our contributions are split into two parts: theoretical and empirical. Our **theoretical contributions** include introducing an algorithm (similar in form to that of Dasgupta et al. and Cicalese et al.) and proving novel lower bounds to show that **our algorithm is optimal up to constants for the returned subset size and optimal up to extraneous log factors for the number of learner queries**. We furthermore provide a framework that justifies the use of “error-squashing” which is used by our work, Cicalese et al., and Dasgupta et al. (present in their code).

Our **empirical contribution** involves a *significantly* higher quality evaluation than prior work. In particular, Dasgupta et al. includes **no baselines** and Cicalese et al. includes **only random sampling** as a baseline, not even the algorithm from the earlier work of Dasgupta et al. We furthermore study six non-synthetic datasets, three different models, and the effect of model transfer on subset selection. In comparison, Cicalese et al. evaluate on a variety of mostly tabular datasets which are much less familiar (with the exception of MNIST) to the data selection literature, with decision-tree-based models (also much less familiar to the data selection literature), and no subset transfer experiments. Dasgupta et al. evaluate only on MNIST (with a non-standard, low-performing model) and FMNIST (with a very small CNN).

Note that we do not claim that our algorithm is empirically stronger than the other machine teaching algorithms, but instead show that the three machine teaching algorithms have approximately equal and state-of-the-art empirical performance. Perhaps the largest contribution of this work is **highlighting the empirical effectiveness of machine teaching methods** so that they can be built upon or used for inspiration in the coreset selection and data pruning literatures.

---

### Decision · Program_Chairs · 2023-01-20

**Decision:**

Reject

**Justification For Why Not Higher Score:**

The paper requires a major revision and thereby another round of peer review.

**Justification For Why Not Lower Score:**

N/A

**Metareview: Summary, Strengths And Weaknesses:**

There is a consensus among the expert reviewers that this paper provides interesting theoretical and experimental analyses of a machine teaching algorithm for data subset selection. The paper provides novel lower bounds and extensive empirical studies that demonstrate the effectiveness of machine teaching methods for data subset selection.

Despite praise for the quality of theoretical and empirical results, a majority of reviewers still have reservations about the novelty of this work, pointing out that the contribution is incremental with respect to that of Dasgupta et al. and Cicalese et al. In particular, one of the reviewers highlighted that the novelty of this work has not been framed appropriately with respect to previous work.

As a result, I cannot recommend the paper in its current form for publication at ICLR 2023 as it does not meet the acceptance bar of the conference.

**Summary Of Ac-Reviewer Meeting:**

N/A